# A unique chromatin profile defines adaptive genomic regions in a fungal plant pathogen

David E Cook[1,2]*, H Martin Kramer[2], David E Torres[2,3], Michael F Seidl[2,3], Bart P H J Thomma[2,4]*

[1]Department of Plant Pathology, Kansas State University, Manhattan, United States; [2]Laboratory of Phytopathology, Wageningen University & Research, Wageningen, Netherlands; [3]Theoretical Biology & Bioinformatics Group, Department of Biology, Utrecht University, Utrecht, Netherlands; [4]University of Cologne, Institute for Plant Sciences, Cluster of Excellence on Plant Sciences (CEPLAS), Cologne, Germany

**Abstract** Genomes store information at scales beyond the linear nucleotide sequence, which impacts genome function at the level of an individual, while influences on populations and long-term genome function remains unclear. Here, we addressed how physical and chemical DNA characteristics influence genome evolution in the plant pathogenic fungus *Verticillium dahliae*. We identified incomplete DNA methylation of repetitive elements, associated with specific genomic compartments originally defined as Lineage-Specific (LS) regions that contain genes involved in host adaptation. Further chromatin characterization revealed associations with features such as H3 Lys-27 methylated histones (H3K27me3) and accessible DNA. Machine learning trained on chromatin data identified twice as much LS DNA as previously recognized, which was validated through orthogonal analysis, and we propose to refer to this DNA as adaptive genomic regions. Our results provide evidence that specific chromatin profiles define adaptive genomic regions, and highlight how different epigenetic factors contribute to the organization of these regions.

*For correspondence:
decook@ksu.edu (DEC);
bart.thomma@wur.nl (BPHJT)

**Competing interests:** The authors declare that no competing interests exist.

## Introduction

Genomes are not randomly organized and comprise complex information beyond their linear nucleic acid sequence (*Sexton and Cavalli, 2015*). While scientific understanding of genome biology continues to grow, significant efforts in the past decade have focused on sequencing new species and additional genotypes of those species (*David et al., 2019*). However, there is a great need to decode the complex information stored in these genomes, to understand genomic responses over various time scales, and ultimately to more fully understand how genotypes lead to phenotypes. With the growing number of high-quality, highly contiguous genome assemblies it is possible to analyze genome organization into chromosomes at high resolution (*Thomma et al., 2016*). Present day genome organization reflects evolutionary solutions to the challenges of information processing and adaptation; a genome must faithfully pass vast amounts of information across cell cycles and reproduction, packaged into limited physical space, while achieving correct access to the information in response to developmental, environmental, or chemical signals. In addition, there needs to be appreciable stochastic genetic variation to ensure that phenotypic variation is present for unknown future events. Organisms undergoing mainly asexual reproduction face an additional evolutionary constraint as they must generate this genetic variation in the absence of meiotic recombination (*Seidl and Thomma, 2014*). Many economically important fungal plant pathogens are either asexual or undergo more frequent asexual reproduction compared to sexual reproduction (*Giraud et al., 2010*). Interestingly, fungal pathogens are subject to additional evolutionary pressure from their

hosts, as host-pathogen interactions create dynamical systems with shifting, yet near-constant selective pressure on the two genomes (*Jones and Dangl, 2006*). These attributes make plant-fungal interactions a particularly interesting system to study aspects of genome evolution and genome organization (*Raffaele and Kamoun, 2012*; *Möller and Stukenbrock, 2017*).

Plant invading microbes use effectors to suppress, avoid, or mitigate the plant immune system (*Oliveira-Garcia and Valent, 2015*; *Cook et al., 2015*). Plants in-turn use a variety of plasma-membrane-bound and cytoplasmic receptors to recognize invasion, through recognition of the effector or its biochemical activity, creating a strong selective pressure on the microbe to modify the effector or its function to alleviate recognition (*Couto and Zipfel, 2016*; *Liang and Zhou, 2018*). The plant pathogenic fungus *Verticillium dahliae* causes vascular wilt diseases on hundreds of plant hosts. *V. dahliae* is presumed asexual and generates genomic diversity in the absence of sexual recombination through large-scale chromosome rearrangements and segmental duplications (*Klosterman et al., 2011*; *de Jonge et al., 2013*; *Faino et al., 2016*; *Shi-Kunne et al., 2018*). The regions undergoing such duplications and rearrangements are hypervariable between *V. dahliae* isolates, and consequently have been referred to as Lineage-Specific (LS) regions. These LS regions are often formally defined based on presence/absence variation (PAV) between strains of a species, and regions that are not designated LS are considered core. These LS regions are enriched for in planta expressed genes and harbor many effector genes contributing to host infection (*de Jonge et al., 2013*; *de Jonge et al., 2012*; *Kombrink et al., 2017*). Similar non-random genomic arrangement of effectors have been reported across diverse plant pathogenic fungal and oomycete genomes (*de Jonge et al., 2013*; *Ma et al., 2010*; *Raffaele et al., 2010*; *Rouxel et al., 2011*; *Goodwin et al., 2011*; *Dutheil et al., 2016*; *Tsushima et al., 2019*; *Peng et al., 2019*). One summary of these observations is referred to as the two-speed genome, in which repeat-rich regions harboring effectors evolve more rapidly than genes outside these regions (*Dong et al., 2015*).

Previous research in various plant-associated fungi has established a link between posttranslational histone modifications and transcriptional regulation of adaptive trait genes. These genes include effectors that facilitate host infection and secondary metabolite (SM) clusters that code for genes that produce chemicals important for niche fitness (*Macheleidt et al., 2016*). By removing or reducing enzymes responsible for particular repressive histone modifications, such as di- and trimethylation of Lys9 and Lys27 residues of histone H3 (H3K9me2/3 and H3K27me2/3), a disproportionally high number of effector and SM cluster genes are derepressed, although a direct role of these marks in transcriptional control was not demonstrated (*Soyer et al., 2014*; *Connolly et al., 2013*; *Studt et al., 2016*). However, evidence from the fungus *Epichloë festucae* that forms a mutualistic interaction with its grass host *Lolium perenne* indicates that direct transcriptional regulation through histone modification dynamics is possible (*Chujo and Scott, 2014*). Although there are clear indications that the epigenome (i.e. heritable chemical modifications to DNA and histones not affecting the genetic sequence) plays a role in adaptive gene regulation, additional evidence is needed to fully understand this phenomenon.

Epigenetic modifications influence chromatin structure, defined as the DNA-RNA-protein interactions giving DNA physical structure in the nucleus (*Sexton et al., 2012*; *Riddle et al., 2011*). This physical structure affects how DNA is organized in the nucleus and DNA accessibility. Methylation of H3K9 and H3K27 are hallmarks of heterochromatin; DNA that is tightly compacted in the nucleus (*Rea et al., 2000*; *Cao et al., 2002*; *Margueron and Reinberg, 2011*; *Janssen et al., 2018*). H3K9 methylation is not only associated with controlling constitutive heterochromatin, but also tightly linked with DNA cytosine methylation (mC), which serves as an epigenetic mark contributing to transcriptional silencing (*Tamaru and Selker, 2001*). A single DNA methyltransferase gene, termed *Dim2*, performs cytosine DNA methylation in the saprophytic fungus *Neurospora crassa* (*Kouzminova and Selker, 2001*). Histone methylation at H3K9 directs DNA methylation by DIM2 through another protein, termed heterochromatin protein 1 (HP1), which physically associates with both DIM2 and H3K9me3 (*Freitag et al., 2004*; *Honda and Selker, 2008*). Some fungi possess a unique pathway to limit the expansion of repetitive DNA such as transposable elements (TEs) through repeat-induced point mutation (RIP), a mechanism that specifically mutates repetitive DNA in the genome during meiosis and induces heterochromatin formation (*Freitag et al., 2002*; *Lewis et al., 2009*). The mutations occur at methylated cytosines resulting in conversion to thymine (C to T mutation) (*Selker et al., 2003*). H3K27 methylation is associated with heterochromatin that is thought to be more flexible in its chromatin status and exist as bivalent chromatin that may be either

transcriptionally repressed or active depending on developmental stage or environmental cues (*Ernst et al., 2011*; *Bemer and Grossniklaus, 2012*; *Gaydos et al., 2014*; *Dattani et al., 2018*). The deposition of H3K27me3 is controlled by a histone methyltransferase that is a member of a complex of proteins termed Polycomb Repressive Complex 2 (PRC2), with orthologs of the core machinery present across many eukaryotes (*Margueron and Reinberg, 2011*; *Freitag, 2017*).

In addition to the role of heterochromatin in transcriptional regulation in filamentous fungi, epigenetic marks contributing to heterochromatin also influence genome organization, and may in tun influence evolution (*Seidl et al., 2016*). For example, H3K9me2/3 is required in *Schizosaccharomyces pombe* and *N. crassa* to form functional centromeres (*Folco et al., 2008*; *Smith et al., 2011*), and H3K9me2/3 is enriched at centromeres of *Cryptococcus neoformans*, *Fusarium graminearum*, and *V. dahliae* (*Dumesic et al., 2015*; *Smith et al., 2012*; *Seidl et al., 2020*). Interestingly, centromeres are diverse genomic regions across fungi, and likely contribute to karyotype variation of related species (*Seidl et al., 2020*; *Sankaranarayanan et al., 2020*). Heterochromatin also influences the physical arrangement of DNA in the nucleus, with strong inter- and intra-heterochromatin DNA-DNA interactions reported in *N. crassa* (*Galazka et al., 2016*; *Klocko et al., 2016*). Recent experimental evidence in *Zymoseptoria tritici*, a fungal pathogen of wheat, indicates that another histone modification associated with heterochromatin, H3K27me3, promotes genomic instability (*Möller et al., 2019*). In the oomycete plant pathogens *Phytophthora infestans* and *Phytophthora sojae* a clear association exists between gene-sparse and transposon-rich regions of the genome and the occurrence of adenine N6-methylation (6mA) (*Chen et al., 2018*). Collectively, these examples point toward unexplained connections between the epigenome, genome organization, and genomic variability, potentially with adaptive consequences. To examine the hypothesis that epigenetic modifications influence the adaptive LS regions of *V. dahliae*, we performed a series of genetic, genomic, and machine learning analyses to characterize these regions in greater detail.

## Results

### DNA cytosine methylation occurs at TEs

To understand the role of DNA methylation in *V. dahliae*, whole-genome bisulfite sequencing, in which unmethylated cytosine bases are converted to uracil while methylated cytosines remain unchanged (*Clark et al., 1994*; *Lister and Ecker, 2009*), was performed in the wild-type and a heterochromatin protein one deletion mutant (Δ*hp1*). The overall level of DNA methylation in *V. dahliae* is low, with an average weighted methylation percentage (calculated as the number of reads supporting methylation over the number of cytosines sequenced) at CG dinucleotides of 0.4% (*Table 1*). The fractional CG methylation level (calculated as the number of cytosine positions with a methylated read over all cytosine positions) is higher, averaged to 9.7% over 10 kb windows. Weighted and fractional cytosine methylation (mC) levels are statistically significantly higher in the WT compared to the Δ*hp1* mutant for all cytosine contexts (Mann-Whitney U-test and Holm correction, p-value<2.2e-16, *Table 1*; *Figure 1—figure supplement 1*). This result is consistent with the requirement of HP1 for DNA methylation in *N. crassa* (*Freitag et al., 2004*). To understand DNA methylation in the context of the functional genome, DNA methylation was analyzed over genes, promoters, and

**Table 1.** Summary of DNA methylation in *Verticillium dahliae* wild-type (WT) and heterochromatin protein one deletion mutant (Δ*hp1*) as measured by whole genome bisulfite sequencing calculated over 10 kb non-overlapping windows.

| Genotype | Avg. weighted mCG* | Avg. weighted mCHG* | Avg. weighted mCHH* | Avg. fraction mCG* | Avg. fraction mCHG* | Avg. fraction mCHH* |
|---|---|---|---|---|---|---|
| WT | 0.0040 | 0.0037 | 0.0034 | 0.097 | 0.097 | 0.088 |
| Δ*hp1* | 0.0030 | 0.0030 | 0.0032 | 0.082 | 0.083 | 0.079 |

Avg. Weighted, The average of total methylated cytosines in a given context divided by total cytosines in that context in a 10 kb windows; Avg. Fraction, The total cytosine positions with a read supporting methylation divided by total cytosines in a specific context in a 10 kb window; mCG, methylated cytosine residing next to a guanine; mCHG, methylated cytosine residing next to any base that is not a guanine next to a guanine; mCHH, methylated cytosine residing next to any two bases that are not a guanines. *, values are significantly different (p-value<0.001), Mann-Whitney U-test. The distribution of values and p-values for individual comparisons are shown in *Figure 1—figure supplement 1*.

TE. Despite statistically significant differences between WT and Δ*hp1* for gene and promoter methylation, the bisulfite sequencing data shows virtually no DNA methylation at these two features (*Figure 1A*, Mann-Whitney U-test and Holm correction, p-values listed with figure). We attribute the difference to a marginal set of elements having a real difference between the genotypes, but the biological significance is likely negligible (*Figure 1—figure supplement 2*). In contrast, there is a higher degree of methylation at TEs, and weighted mCG is five times higher in the wild-type versus Δ*hp1* (*Figure 1A*, bottom panel). To support this, 31% (697/2227) of the TEs in the wild-type strain have a weighted mCG value greater than or equal to the mean weighted CG methylation level (0.015), while this number is 1% (33/2227) in Δ*hp1*. In contrast, 0% of wild-type (1/11430) and Δ*hp1* (0/11430) genes have a weighted mCG value equal to this level. Notably, the Δ*hp1* strain displays a

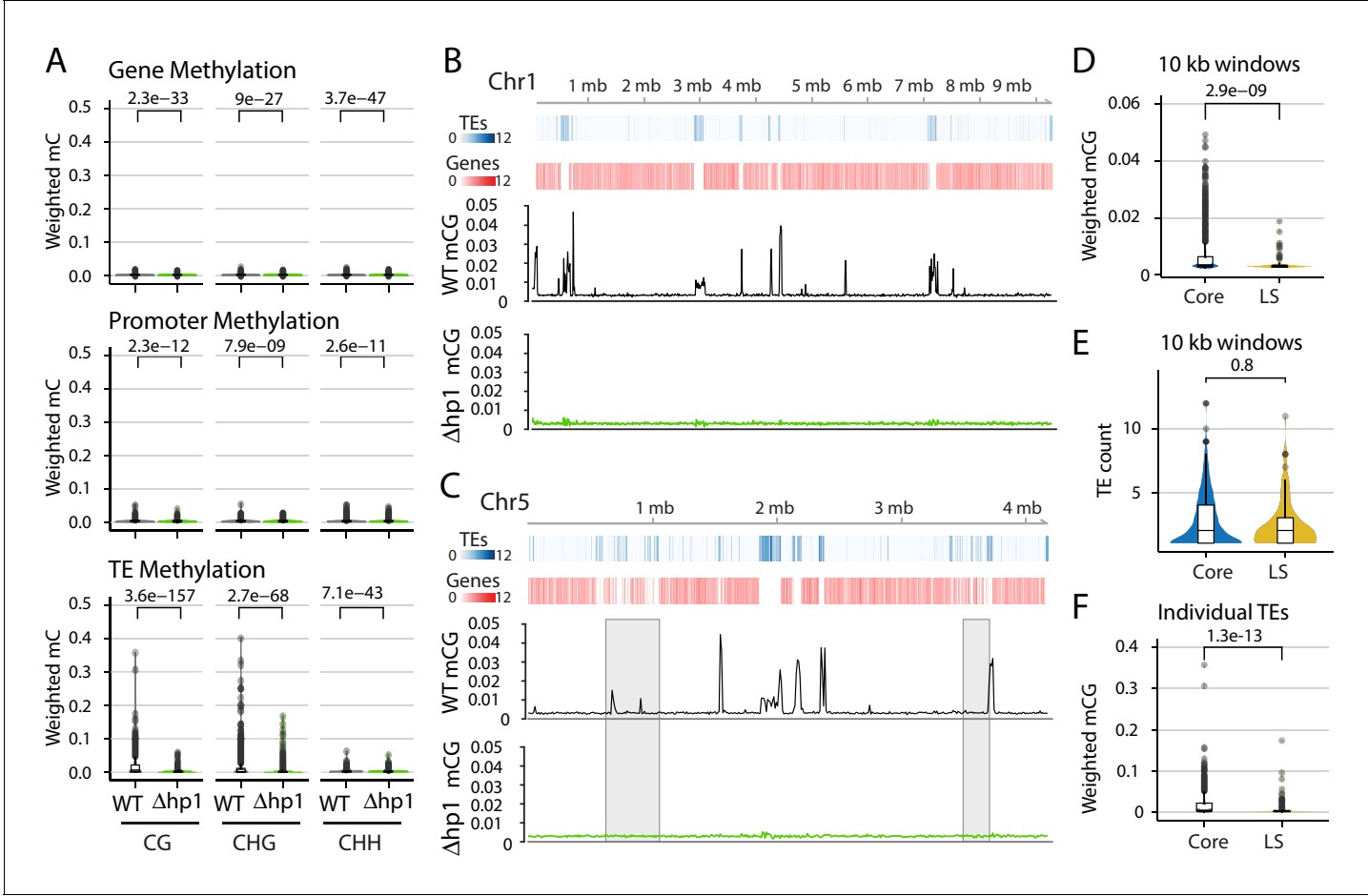

**Figure 1.** DNA methylation is only present at transposable elements, but not at those present in Lineage-Specific (LS) regions. (A) Violin plot of the distribution of DNA methylation levels quantified as weighted methylation over genes, promoters, and transposable elements (TEs). Cytosine methylation was analyzed in the CG, CHG, and CHH sequence context. Methylation was measured in the wild-type (WT) and heterochromatin protein one knockout strain (Δ*hp1*). (B, C) Whole chromosome plots showing TE and Gene counts (blue and red heatmaps) and wild-type (black lines) and Δ*hp1* (green line) CG methylation as measured with bisulfite sequencing. Data is computed in 10 kilobase non-overlapping windows. (C) Two previously defined LS regions (*Faino et al., 2016*) are highlighted by gray windows. (D) Violin plot of weighted cytosine methylation in 10 kb windows broken into core versus LS location (E) Same as D but plots are for the counts of TEs per 10 kb window. (F) Same as in D but methylation levels were computed at individual TE elements. (A,D,E,F) Statistical differences for indicated comparisons were carried out using non-parametric Mann-Whitney U-test and Holm multiple testing correction with associated p-values shown.

The online version of this article includes the following source data and figure supplement(s) for figure 1:

**Source data 1.** Genome-wide cytosine methylation levels in 10 kb windows in wild-type and Δhp1 Verticillium dahliae.

**Figure supplement 1.** Genome-wide cytosine methylation in wild-type and Dhp1.

**Figure supplement 2.** Cytosine methylation for functional elements in wild-type and Dhp1.

**Figure supplement 3.** Transcriptional impact of Dhp1.

reduced growth rate, altered colony morphology, reduced spore production, and reduced virulence, likely attributable to the pleiotropic effects of genome-wide heterochromatin alterations (*Kramer et al., 2020*).

To further analyze DNA methylation levels and confirm that the low DNA methylation levels in the wild-type strain are indeed different than those in Δ*hp1,* CG DNA methylation levels were plotted in 10 kb windows across individual chromosomes. These plots clearly show that DNA methylation is not continuously present across the *V. dahliae* genome, and DNA methylation is significantly different between wild-type and Δ*hp1* (*Figure 1B,C*). Furthermore, regions in the genome with higher densities of TEs and lower gene numbers have higher levels of DNA methylation, consistent with the global DNA methylation summary (*Figure 1B,C*). Interestingly, these results show that while DNA methylation is only present at TEs, not all TEs are methylated, a phenomenon that was previously described as 'non-exhaustive' DNA methylation (*Montanini et al., 2014*). To further understand this phenomenon, we sought to identify discriminating genomic features that could account for some TEs not being methylated. The whole-chromosome methylation data suggested a lack of DNA methylation at previously identified LS regions (*Figure 1C*, gray windows). These LS regions were previously detailed for *V. dahliae,* and are characterized as regions that are highly variable between isolates of the species, are enriched for actively transcribed TEs, and contain an increased proportion of genes involved in host virulence (*Klosterman et al., 2011*; *de Jonge et al., 2013*; *Faino et al., 2016*). Thus, we tested if DNA sequences at LS regions are less frequently methylated by comparing weighted mCG levels in 10 kb bins containing at least one TE for core versus LS regions. This analysis showed significantly more DNA methylation for core bins (Mann-Whitney U-test and Holm correction, p-value=2.9e-9, *Figure 1D*), which cannot be accounted for by a simple difference in the number of TEs in the core and LS regions analyzed (Mann-Whitney U-test and Holm correction, p-value=0.8, *Figure 1E*). Higher CG methylation levels also hold true when analyzed at the level of individual TE elements (Mann-Whitney U-test and Holm correction, p-value=1.3e-13, *Figure 1F*). Analyzing the transcriptional impact of Δ*hp1* on in vitro grown cultures identified 1522 genes that were expressed significantly higher in Δ*hp1* compared to wild-type (log2 fold-change >1 and adjusted p-value<0.01, *Figure 1—figure supplement 3A*). Of the genes more highly expressed in Δ*hp1*, those located in LS regions showed a greater transcriptional increase (*Figure 1—figure supplement 3B*). However, as we have shown that loss of DNA methylation alone does not result in altered gene transcription (*Kramer et al., 2020*), the differential gene expression caused by *HP1* mutation is likely caused by additional effects on chromatin and DNA compaction (*Jamieson et al., 2016*). Collectively, these analyses demonstrate that DNA methylation occurs almost exclusively at TEs and, importantly, that not all TEs are methylated. This observation can in part be explained by mCG differences for TEs in the core versus LS regions.

## TE classes have distinct genomic and epigenomic profiles

To understand the functional status of the various TEs in the genome, DNA-histone modification location data were collected using chromatin immunoprecipitation followed by sequencing (ChIP-seq) against H3K9me3 and H3K27me3, which allows for the identification of DNA interacting with these modified histones. Characteristics of TE sequence, such as GC percentage, composite RIP index (CRI), and TE age, estimated as the Jukes-Cantor distance to the consensus sequence of the specific TE family, were calculated (see Materials and methods). To further classify genomic regions as eu- or heterochromatic, we performed an assay for transposase accessible chromatin and sequencing (ATAC-seq) (*Buenrostro et al., 2013*). This method uses a TN5 transposase to restrict physically accessible DNA in the nucleus and tags the DNA ends with oligonucleotides for downstream sequencing. Transcriptional activity was assayed using RNA-sequencing. To analyze all of these TE characteristics (variables) at once, dimensional reduction with principal component analysis (PCA) was employed, which facilitates data interpretation on two-dimensions to identify important variables and their relationships within large datasets. The individual TEs were grouped into four broad classes (Type I DNA elements and Type II LTR, LINEs, and Unspecified elements) and analyzed for each measured variable (*Supplementary file 1*- table 1). The first dimension of PCA shows the largest separation of the data points and variables, and largely separates the data based on euchromatin versus heterochromatin features (*Figure 2A*, PC1). This is seen by the variables ATAC-seq, % GC, RNA-sequencing, H3K9me3 ChIP, CRI, and DNA methylation (mCG) being furthest separated along the x-axis (*Figure 2A*). Open chromatin features such as higher ATAC-seq, %GC, and

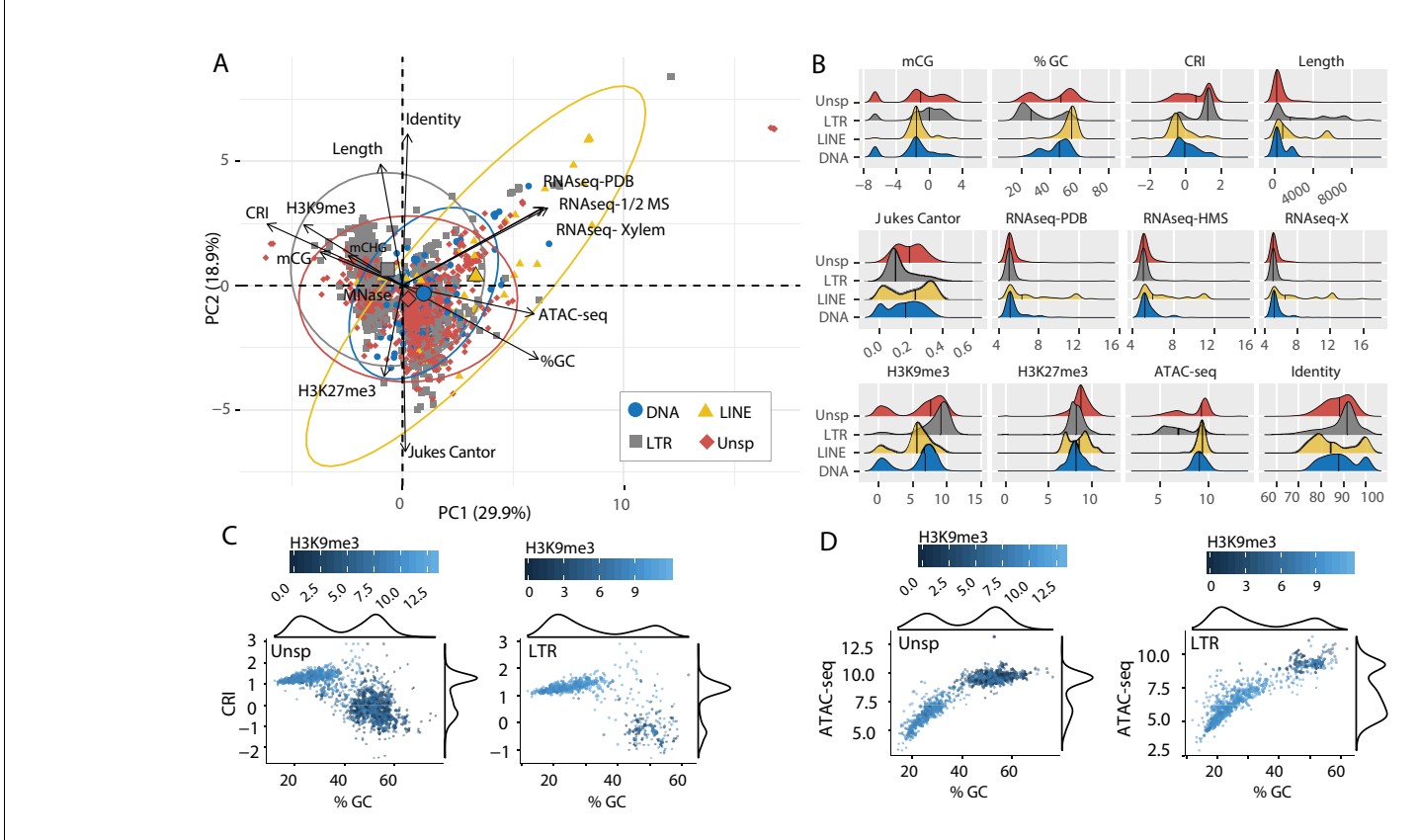

**Figure 2.** Individual TE families have distinct epigenetic and physical compaction profiles. (**A**) Principle component analysis for 14 variables measured for each individual transposable element (TE). Each vector represents one variable, with the length signifying the importance of the variable in the dimension. The relationship between variables can be determined by the angle connecting two vectors. For angles < 90°, the two variables are correlated, while those >90° are negatively correlated. Each individual element is shown and highlighted by color and symbol as indicated by the key. Colored ellipses show the confidence interval for the four families along with a single large symbol to show the mean position for the four families. mCG, weighted CG DNA methylation; mCHG, weighted CHG DNA methylation; CRI, Composite RIP index; %GC, percent GC sequence content; Identity, Nucleotide identity as percent identity to the consensus TE sequence of a family; Length, element length; Jukes Cantor, Jukes Cantor corrected distance as proxy of TE age; RNAseq, RNA-sequencing reads from (PDB), half strength MS (HMS) or tomato xylem sap (Xylem) grown fungus expressed as variance stabilizing transformed log2 values (see Materials and methods for details); H3K9me3, log2 (TPM+1) values of mapped reads from H3K9me3 ChIP-seq; H3K27me3, log2 (TPM+1) values of mapped reads from H3K27me3 ChIP-seq; ATAC-seq, log2 (TPM+1) values for mapped reads from Assay for transposase accessible chromatin. (**B**) Ridge plots showing the distribution of the individual TE families per variable. The median value is shown as a solid black line in each ridge. Variables same as in A except for mCG, log2(weighted cytosine DNA methylation + 0.01). (**C**) Scatter plot for %GC versus CRI values for individual TE elements shown as points. The two plots are for TEs characterized as Unspecified (Unsp) or LTR, labeled in the upper left corner. Each point is colored according to log2 (TPM+1) values from H3K9me3 ChIP-seq, scale shown above each plot. A density plot is shown for both variables on the opposite side from the labeled axis. (**D**) Same as in C, but the y-axis is now showing the log2 (TPM+1) values from ATAC-seq.

The online version of this article includes the following source data and figure supplement(s) for figure 2:

**Source data 1.** Genomic data for transposable elements.

**Figure supplement 1.** Genomic distribution of DNA characteristics by transposable element (TE) classes.

**Figure supplement 2.** Characterization of transposable elements (TEs) in nine subclasses across genomic variables.

**Figure supplement 3.** The LTR subclass distinction does not account for the bimodal distribution of LTR elements in the genome.

transcriptional activity are positive on the x-axis, with small angles between the vectors, indicating correlation among those variables. Conversely, features associated with heterochromatin, such as H3K9me3 association, DNA methylation and indication of RIP (CRI) are all negative on the x-axis, and the position of their vectors indicates correlation among these variables, and negative correlation to the euchromatin features (*Figure 2A*). The second axis discriminates elements based on their H3K27me3 profile and sequence characteristics such as Jukes Cantor (TE age), Identity and Length

(*Figure 2A*). Regarding the TE classifications, there is a stronger association for the LTR and Unspecified elements with the heterochromatin features (*Figure 2A*, gray and red ellipse extending along negative x-axis). Collectively, this multivariate description of TEs identifies those that are more transcribed and open as having lower association with H3K9me3, mCG, and RIP mutation. There are statistically significant differences between the TE types for each of these variables (*Supplementary file 1*- table 2, Dunns test with Benjamini-Hochberg testing correction), and the LTR elements have the highest levels of H3K9me3 and mCG, along with the highest CRI values and lowest %GC, consistent with the mechanistic link between the four variables (*Figure 2B*). Interestingly, a bimodal distribution occurs for %GC and CRI within the LTR and Unspecified elements, indicating that some of the LTR elements have undergone RIP and are heterochromatic, while other elements have not been subject to this mechanism (*Figure 2B*). This delineation occurs for the Unspecified and LTR elements with a %GC sequence content less than approximately 40%, which have positive CRI values and high H3K9me3 signal (*Figure 2C*). A similar distinction is seen with ATAC-seq data that show a clear break around 40% GC content, and elements below this have lower ATAC-seq signal and higher H3K9me3 signal (*Figure 2D*). These trends are not observed for the LINE and DNA elements (*Figure 2—figure supplement 1*). The TEs were further divided into sub-classes (*Supplementary file 1*- table 3), we do not detect an obvious difference to explain these data. For instance, while there is a significant difference between the nine sub-classes for %GC and CRI (Kruskal-Wallis statistic = 382.4, p-value=1.07e-77 %GC; Kruskal-Wallis statistic = 259.41, p-value=1.74e-51 CRI) (*Supplementary file 1*- table 4) the two LTR subclasses (Copia and Gypsy) do not have statistically different means (Conovers test and BH correction, p-value=1 %GC; p-value=0.19 CRI) (*Figure 2—figure supplement 2*; *Supplementary file 1*, table 5). The two LTR subclasses are significantly different for their association with H3K9me3 and H3K27me3 (Conover test and BH correction, p-value=1.03e-06 H3K9me3; p-value=1.08e-10 H3K27me3) (*Figure 2—figure supplement 2*; *Supplementary file 1*- table 5), but the subclasses of LTRs both occur in the two bimodal states, and cannot account for the difference (*Figure 2—figure supplement 3*). It is not possible to distinguish the unspecified elements into reliable subclasses. These results suggest that the most prominent TE classes in the genome, LTR and Unspecified elements, exist in two distinct chromatin states, which we cannot easily distinguish through sequences of sub-classes.

## TE location significantly influences the epigenetic and DNA accessibility profile

To further dissect the relationship between epigenetic modifications, chromatin status and genomic location, pair-wise comparisons were made for all TEs in core versus LS regions. All measured variables, except TE length, are significantly different for TEs in the core versus LS regions (*Figure 3—figure supplement 1*). Further division of the TEs indicated that the LTR and Unspecified elements showed the greatest differences for core versus LS measurements (*Figure 3A*), demonstrating that the major driver of core versus LS differences are driven by the LTR and Unspecified elements. The bimodal distribution for GC content, CRI, H3K9me3, and ATAC-seq can be accounted for in part by core versus LS separation (*Figure 3B*, red versus gray). Collectively, the status of the LS TE elements can be characterized as devoid of DNA and H3K9 methylation, low RIP signal, generally higher than 0.5 GC content, higher levels of H3K27me3, more open with ATAC-seq signal, and higher transcription levels (*Figure 3D*). The core versus LS location is not sufficient to fully explain the chromatin status, as there are many elements located in the core genome that share a similar profile with the LS elements (*Figure 3D*, elements highlighted in black boxes), but as an ensemble, the core elements are statistically different than those found at LS regions.

## Significantly different chromatin status between core and LS regions extends to larger DNA segments

The analysis of TEs in the genome clearly shows that a subpopulation of elements that occur in the previously defined LS regions have different epigenetic modifications and physical openness compared to those in the core genome. LS regions are significant for *V. dahliae* biology as they code many proteins which support host infection (*de Jonge et al., 2013*; *Faino et al., 2016*). To generate a global analysis beyond an analysis of TEs, the genome was analyzed over 10 kb non-overlapping windows. This analysis showed that regions with high TE density tend to have lower GC content,

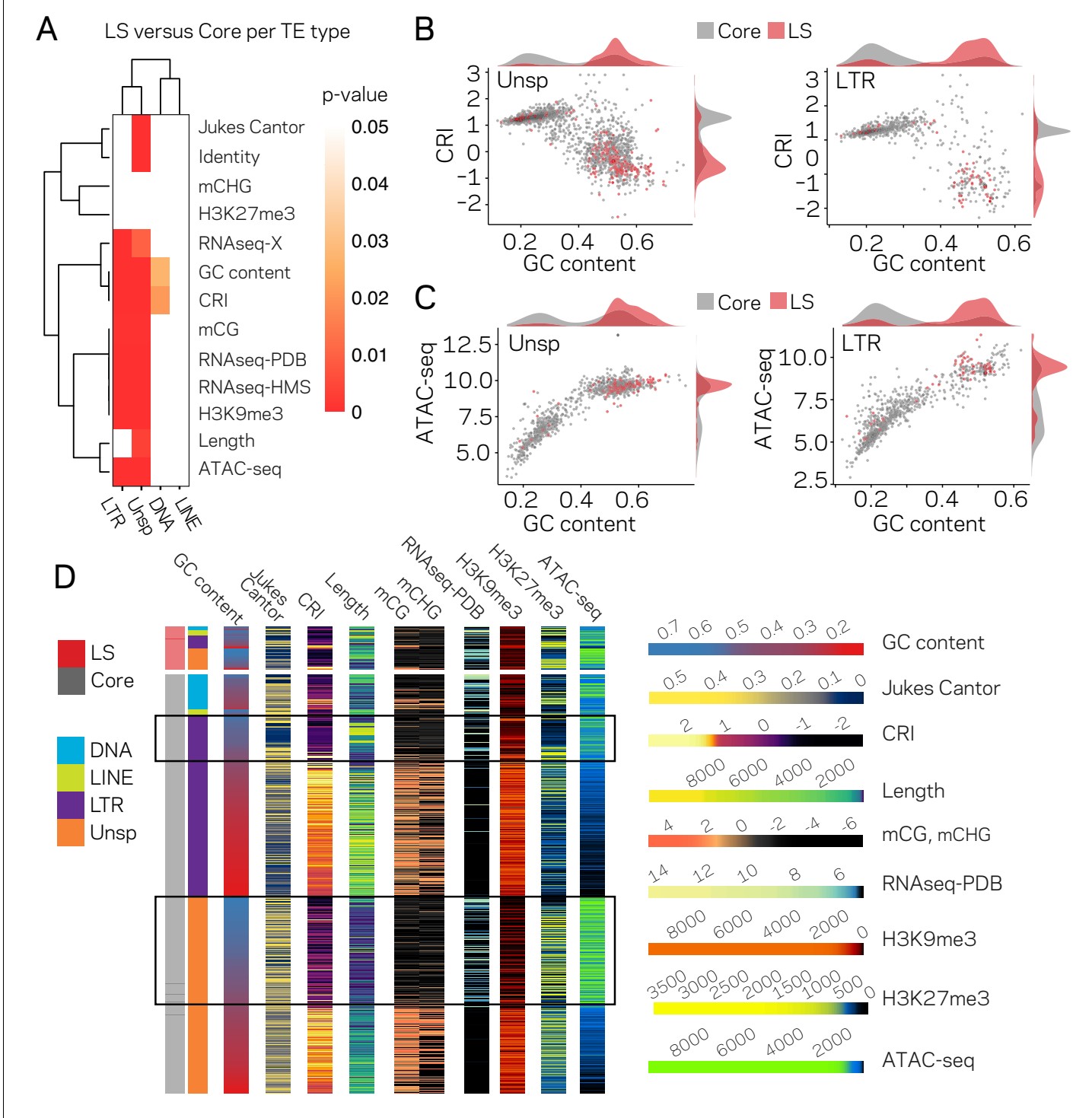

**Figure 3.** The LTR and Unspecified elements have significantly different chromatin profiles based on core versus LS location. (**A**) Heatmap comparing core versus LS values within the four TE classifications for the variable listed to the right. Plot colored based on p-values from Wilcoxon rank sum test. p-values≥0.05 are colored white going to red for p-value ≅ 0. (**B**) Scatter and density plots similar to those shown in *Figure 2c* except the individual TE points are colored by core (gray) versus LS (red) location. The density plots are also constructed based on the two groupings (**C**) Similar to B, with the y-axis now showing the log2 (TPM+1) values from ATAC-seq (**D**) Multiple grouped heatmaps for ten variables collected for each TE. Each row represents a single element and the same ordering is used across all plots. The LS elements are grouped at the top, indicated by the red bar at the top left, and the core elements are grouped below, indicated by the gray bar at the left. Elements are further grouped by the four classifications indicated by the color code shown to the left. Within each element group, the elements are ordered by descending GC content. The scale for each heatmap is

*Figure 3 continued on next page*

*Figure 3 continued*

shown at the right. GC content, fraction of GC in sequence; Jukes Cantor, corrected distance as proxy of TE age; CRI, Composite RIP index; Length, element length; mCG and mCHG, log2(weighted cytosine DNA methylation+0.01) for CG and CHG, respectively; RNAseq-PDB, variance stabilizing transformed log2 RNA-sequencing reads from PDB grown fungus; H3K9me3 and H3K27me3 and ATAC-seq, TPM values of mapped reads H3K9me3 ChIP-seq, H3K27me3 ChIP-seq, or Assay for transposase accessible chromatin, respectively. Black boxes highlight LTR and Unsp elements in the core that have euchromatin profiles.

The online version of this article includes the following figure supplement(s) for figure 3:

**Figure supplement 1.** Violin plots for twelve measured variables collected for the TEs located in either the core (blue) or LS (yellow) regions of the genome.

higher DNA and H3K9 methylation and a lack of ATAC-seq reads. The distribution of H3K27me3 overlaps with regions having DNA and H3K9 methylation, but additional H3K27me3 regions occur that lack DNA and H3K9 methylation (*Figure 4A*). It appeared that regions marked by H3K27me3 that lack H3K9me3 have more open DNA than regions with H3K27me3 and H3K9me3 (*Figure 4A*, ATAC). This is apparent for the LS regions that appear to have increased H3K27me3 signal, lack H3K9me3 and have an intermediate level of DNA accessibility (*Figure 4B*, regions marked by gray boxes). PCA was again employed to combine the variables into a single analysis, with the first dimension explaining nearly 60% of the variation in the data (*Figure 4C*). The first dimension largely captures the variables describing euchromatin versus heterochromatin, such that ATAC-seq and % GC are furthest separated on the x-axis from H3K9me3, DNA methylation and TE density (*Figure 4C*). Interestingly, the DNA segments classified as core are mostly separated across the first dimension (*Figure 4C*). The second and third dimensions of the PCA explained a similar amount of variation in the data; 14.4% and 10.7%, respectively. Data from the RNA-seq experiment contributed nearly all the information to the second dimension, while the H3K27me3 ChIP-seq data contributed most of the information in the third dimension (*Figure 4—figure supplement 1*, and *Supplementary file 1*- table 6). Interestingly, when this third dimension is considered, we observe a separation of the core from the LS regions (*Figure 4C*,y-axis). This observation suggests that LS regions are more discernable from the core based on variation for H3K27me3, and less by variation for DNA openness, and DNA or H3K9 methylation.

Our observations can be summarized into a genome-wide model; for the core genome, regions with higher TE density have low ATAC-seq signal (closed DNA) and elevated H3K9me3 signal and thus represent the heterochromatic regions (*Figure 4D*, cluster of large blue dots plotted at middle left). Core genomic regions that are gene-rich have a higher ATAC-seq and lower H3K9me3 signal and represent the euchromatic portion of the genome (*Figure 4D*, cluster of small blue dots plotted in the lower-middle section). The LS regions are a hybrid of the two that contain high TE density and higher H3K27me3 signal but have higher ATAC-seq signals when compared with similar TE containing regions in the core genome (*Figure 4D*, cluster of large yellow triangles plotted in the middle). This simple model of the genome accounts for many of the phenomena described here, and links the epigenome, physical genome and functional genome.

## Machine learning predicts more lineage-specific genomic regions than previously considered

Given that a clear model emerges that links the epigenome and physical openness of DNA with adaptive regions of the genome, we assessed the extent to which these features can predict core or LS regions. Stimulated by our observations (*Figure 4*), we used ATAC-seq, RNA-seq, H3K27me3, TE density, and H3K9me3 along with the binary classification of the 10 kb windows as core or LS for machine learning. Four supervised machine learning algorithms were used to train (i.e. learn) on 80% of the data (2890 regions) using a 10-fold cross validation repeated three times, while the remaining 20% (721 regions) were used to test the model (i.e. predict). Repeated cross-validation and parameter tuning identified the best model parameters based on accuracy (*Figure 5—figure supplement 1*). Assessing the classifier's performance using area under the receiver operating characteristic (auROC) curve suggested excellent results ranging from 0.93 to 0.95 (value of 1 is perfect prediction) (*Figure 5A*). While auROC is the *de facto* standard for machine learning performance (*Bradley, 1997*), it is not appropriate for assessing predictive performance of binary classification

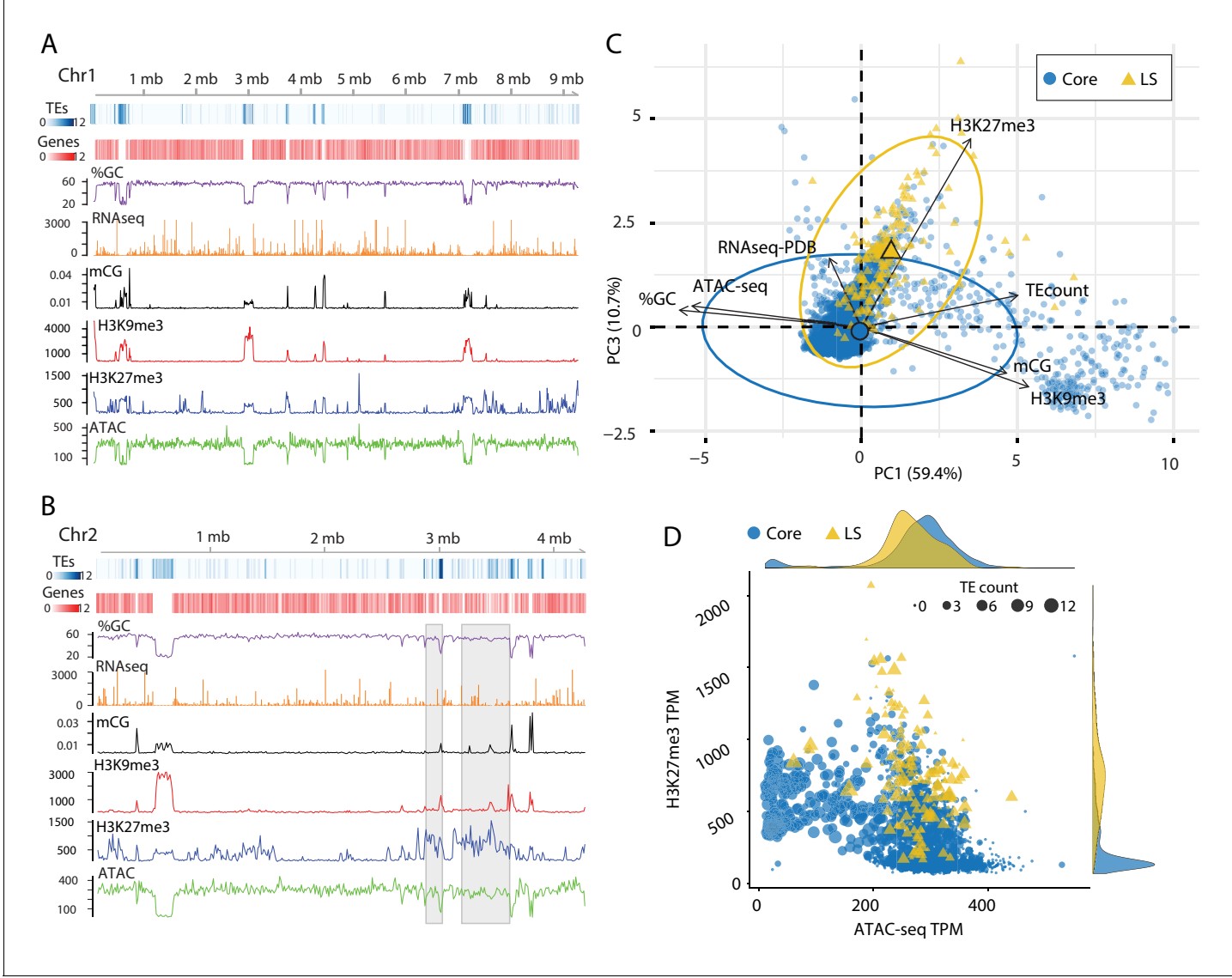

**Figure 4.** Epigenome and physical DNA characteristics collectively define core and LS regions. (A and B) Whole chromosomes plots showing TE and gene counts over 10 kb genomic windows, blue and red heatmaps respectively. The %GC content is shown in purple, RNA-seq show in orange, CG cytosine DNA methylation shown in black, H3K9me3 and H3K27me3 ChIP-seq shown in red and blue respectively, and ATAC-seq shown in green. Values are those previously described. (B) Two LS regions are highlighted with a gray window. (C) Principle component analysis for seven variables at each 10 kb window. Dimensions 1 and 3 are plotted and collective explain ~70% of the variation in the data. The individual symbols are colored by genomic location with core (blue circles) and LS (yellow triangles). Colored ellipses show the confidence interval for the core and LS elements with a single large symbol to show the mean. (D) Scatter plot of the 10 kb windows colored for core and LS location by ATAC-seq data (TPM, x-axis) and H3K27me3 (TPM, y-axis). The size of each symbol is proportional to the TE count, shown as five possible ranges from 0 (smallest), 1–3 (next, larger), up to 10–12 (largest). The density plot of each variable is shown on the opposite axis.

The online version of this article includes the following source data and figure supplement(s) for figure 4:

**Source data 1.** Genomic data for 10 kb windows.

**Figure supplement 1.** Principle component analysis for seven variables genome wide at 10 kb window.

problems when the two classes are heavily skewed as it overestimates performance due to the high number of true negatives (*Davis and Goadrich, 2006*). This is the case for our analysis in which the test set (721 regions) contains only 33 of the known LS regions (4.6%). To more accurately assess model performance, precision-recall curves were employed as these do not use true negatives, and are therefore less influenced by skewed binary classifications (*Saito and Rehmsmeier, 2015*). All

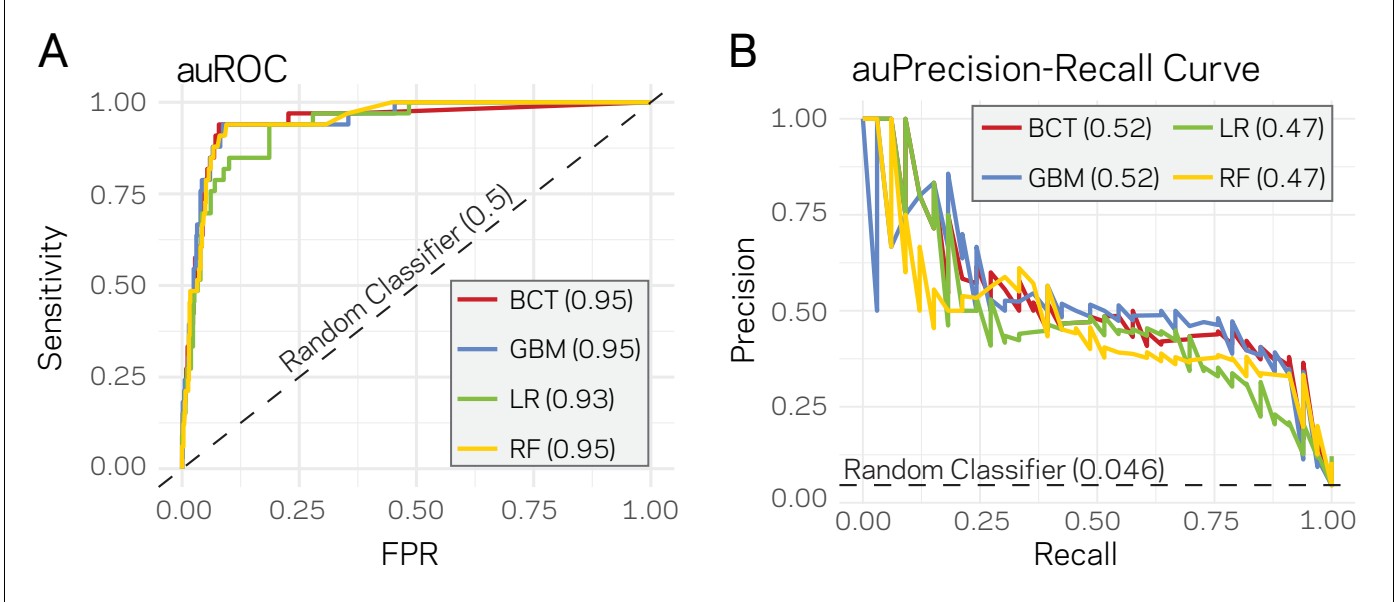

**Figure 5.** Supervised machine learning can predict Lineage-Specific (LS) regions based on epigenome and physical genome characteristics. (**A**) Area under the Response operator curve (auROC) plotting sensitivity and false positive rate (FPR) for four machine learning algorithms, BCT- Boosted classification tree; GBM- stochastic gradient boosting; LR- logistic regression; RF- random forest. The auROC scores are shown next the algorithm key in the gray box. The black dotted line represents the performance of a random classifier. Perfect model performance would be a curve through point (0,1) in the upper left corner. (**B**) Area under the Precision-Recall curve for the same four models shown in A. Area under the curves are shown in the figure key in the gray box. The black dashed line shows the performance of a random classifier, calculated as the TP / (TP + FN). Perfect model performance would be a curve through point (1,1) in the upper right corner.

The online version of this article includes the following figure supplement(s) for figure 5:

**Figure supplement 1.** Results from model parameter tuning and assessment.

four algorithms consistently outperformed a random classifier, with the boosted classification tree (BCT) and stochastic gradient boosting (GBM) algorithms having the same highest area under the precision-recall curve of 0.52 (*Figure 5B*). Each final model was used to classify the test data. The resulting confusion matrix indicated that the BCT model only identified 13 of the 33 LS regions (*Table 2*), resulting in poor recall (*Table 3*). In contrast, the other three models did identify most of the known LS regions (high recall), but had lower precision caused by the high rate of false positives

**Table 2.** Confusion Matrix for LS and core prediction in *V.dahliae* from test data classification using the final trained model.

| | | Known | |
| --- | --- | --- | --- |
| | **Predicted** | **Core** | **LS** |
| LR | Core | 638 | 7 |
| | LS | 50 | 26 |
| GBM | Core | 645 | 5 |
| | LS | 43 | 28 |
| BCT | Core | 672 | 20 |
| | LS | 16 | 13 |
| RF | Core | 623 | 2 |
| | LS | 65 | 31 |

LR, Logistic Regression; GBM, Stochastic Gradient Boosting; BCT, Boosted Classification Tree; RF, Random Forest; Core, regions of the genome defined as core; LS, regions of the genome defined as Lineage Specific. For final model parameter settings and classification thresholds, see Materials and methods.

**Table 3.** Assessment of four trained machine learning algorithms on final test data.

| Models | Precision | Recall | MCC | F1 |
|---|---|---|---|---|
| LR | 0.34 | 0.79 | 0.49 | 0.48 |
| GBM | 0.39 | 0.85 | 0.55 | 0.54 |
| BCT | 0.45 | 0.39 | 0.39 | 0.42 |
| RF | 0.32 | 0.94 | 0.52 | 0.48 |

LR, Logistic Regression; GBM, Stochastic Gradient Boosting; BCT, Boosted Classification Tree; RF, Random Forest; MCC, Matthews Correlation Coefficient; F1, F-score or harmonic mean of precision and recall. For final model parameter settings and classification thresholds, see Materials and methods.

(*Tables 2* and *3*). We also assessed the final model performance using Matthews correlation coefficient (MCC) because it is a better measure of binary classification performance for unbalanced data sets (*Sokolova et al., 2006*). The GBM and random forest (RF) models had the highest MCC values for our experiments (*Table 3*).

Our original intention for running the ML analysis was to determine if genomic and chromatin characteristics could be used to identify the previous LS and core regions, which our results show to be true (e.g. high recall). However, subsequently we were interested to explore the relatively large number of false positives (i.e. regions classified as LS by the ML analysis that were previously classified as core). Since the original classification of core and LS in *V. dahliae* was based on presence/absence variation of only a limited set of strains (*de Jonge et al., 2013*; *Faino et al., 2016*), we reasoned it was possible that the false positives could in fact be genuine LS regions that were originally missed. Alternatively, the false positive regions may be errors by the ML models, and therefore not genuine LS regions.

The two best models from the initial testing, GBM and RF, were used to further understand the nature of the false positives. The GBM and RF models predicted a total of 96 and 81 regions as LS, respectively, suggesting there could be two to three times more LS DNA than previously identified. We re-ran the GBM and RF algorithms on 15 new training-test splits, independently training and predicting on each set (see Materials and methods for details). This approach nearly saturated the genome, providing multiple predictions per window and only 124 of the 3611 regions were missed (*Figure 6—figure supplement 1*). The average MCC performance estimate of the GBM and RF classifiers were 0.53 and 0.48 over the 15 runs, and our results indicate consistent performance across the independent predictions (*Figure 6A*; *Figure 6—figure supplement 2*; *Supplementary file 1*-Tables 7 and 8). The GBM classifier predicted a total of 285 of the 10 kb regions as LS, while the RF classifier predicted 388 (*Supplementary file 1*- table 9,10). The LS predictions for the two models were in agreement for 280 regions, which is 98% of the GBM predictions and 72% of those from the RF (*Figure 6B*), overall showing high agreement between the two classifiers (*Figure 6—figure supplement 3*).

A consensus prediction was generated for each 10 kb DNA segment, where a region was classified LS if predicted by both models, resulting in 280 high confidence LS regions. An additional 41 regions were classified LS using a conservative joining step in which a single predicted core region was called LS if it was flanked by LS predictions on both sides (see Materials and methods). LS regions that were consecutive or within 20 kb of each other were concatenated to form longer LS segments, resulting in a total of 91 LS regions comprised of 3.33 Mb of DNA (*Figure 6C*). The original LS classification liberally merged LS regions to form four main regions and an additional four smaller regions, altogether totaling 8 LS regions and 1.82 Mb of DNA (*Faino et al., 2016*). Comparing the new consensus LS classification with the original classification, we find the new LS regions have a shorter, but not statistically significantly different, mean length (mean length = 36578 bp new LS, mean length = 226898 bp old LS, Mann-Whitney U-test, Holm adjusted p-value=0.93) (*Figure 6—figure supplement 4*), but the total amount of DNA classified as LS has increased nearly two times. Also, the newly defined set of LS regions appears to more clearly delineate the LS and core regions (*Figure 6—figure supplement 5*). That is, using the new core and LS classification, the separation between core and LS becomes more distinct. The newly described LS regions go beyond the original

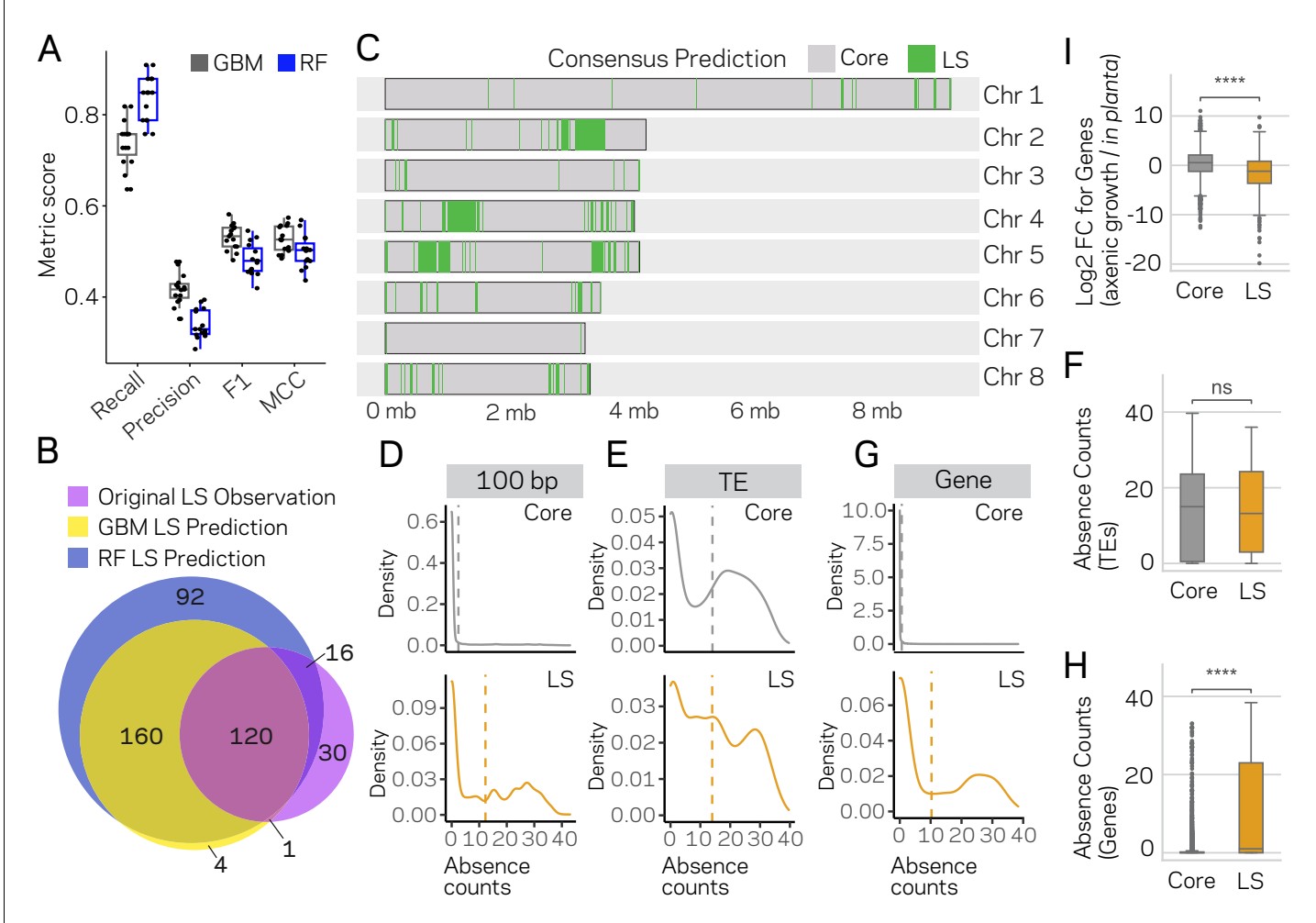

**Figure 6.** Machine learning predictions for genome-wide LS content. (**A**) Two machine learning algorithms, Stochastic Gradient Boosting (GBM) and Random Forest (RF), were used to predict Lineage-Specific (LS) regions from 15 independent training-test splits (80/20). Classifier performance was measured for each of the 15 trials, and summarized as a boxplot with each trial represented as a point. (**B**) Venn diagram showing the overlap between the results of the two classifiers and the original observations of LS regions (*de Jonge et al., 2013*; *Faino et al., 2016*). Each slice of the diagram shows the number of LS regions predicted, see Materials and methods for additional details. (**C**) Schematic representation of the eight chromosomes (labeled on right) of *V. dahliae* strain JR2. Core (gray) and LS (green) classification for 10 kb windows. The consensus predictions were those made by both the GBM and RF model (in total 280). (**D**) Boxplot showing a significant difference for in planta gene induction between core and LS genes, Mann-Whitney U test p-value=1.34e-50. (**E**) Density distribution for core (gray) and LS (orange) elements based on absence counts over 100 bp windows. The mean absence counts are shown as a dashed vertical line. (**F**) Similar to E but the analysis was conducted for TEs. (**G**) Boxplot showing no significant difference between core and LS TE elements for absence counts, Mann-Whitney U test p-value=0.92. (**H**) Similar to E but the analysis was conducted for genes. (**I**) Boxplot showing a significant difference between core and LS genes for absence counts, Mann-Whitney U test p-value=3.82e-104. ns, non-significant; **** p-value<1.00e-4.

The online version of this article includes the following source data and figure supplement(s) for figure 6:

**Source data 1.** Consensus LS classification genomic regions.
**Source data 2.** Gene presence and absence counts.
**Source data 3.** TE presence and absence counts.
**Figure supplement 1.** Density plot for the number of distribution of predictions per genomic region.
**Figure supplement 2.** Recall and Precision assessment for independent classification trials.
**Figure supplement 3.** Genomic location of Lineage-Specific (LS) predictions from two ML models.
**Figure supplement 4.** Size distribution and summary description of the New and Old Lineage-Specific (LS) classifications.
**Figure supplement 5.** Genome model of core and Lineage-Specific (LS) regions defined by epigenetics and chromatin status.

definition of LS (*Klosterman et al., 2011*; *de Jonge et al., 2013*), and as such, we propose to refer to the newly defined LS regions as 'adaptive genomic regions'.

Under the hypothesis that the newly identified classifications from ML were not errors, but represented bona fide LS regions, we expect the new classifications to maintain distinct characteristics between LS and core regions. Namely, LS regions having extensive sequence variability between *V. dahliae* strains (*Klosterman et al., 2011*; *Faino et al., 2016*; *de Jonge et al., 2012*), and they are enriched for in planta induced genes, and genes that are known or presumed to be involved in host infection (*de Jonge et al., 2013*). To this end, we analyzed PAV and performed genome-wide enrichment analyses for in planta-induced genes, for genes coding secreted proteins, and for effector genes in old and new LS regions. To analyze PAV, we summarized missing DNA segments, termed absence counts, from 42 *V. dahliae* strains based on whole-genome sequencing (see Materials and methods for details). The original LS classification was defined by PAV, and thus we anticipated that the updated LS classification, if valid, should still reflect higher variability for LS regions between *V. dahliae* strains. The analysis showed that the majority (82.6%) of 100 bp windows classified as core were present in all 42 strains, and the distribution of absence counts suggested low variation (*Figure 6D*, mean absence count = 2.5). In contrast, only 37.3% of LS classified regions were present in all 42 strains, and had a significantly higher mean absence count of 12.3 (Wilcoxon rank sum, p<2.2e-16) (*Figure 6D*). These results help validate our approach by showing the PAV disparity between LS and core elements is readily observed in this set of 42 strains. We assessed the absence counts for TEs and genes to understand what functional elements account for the differences between the LS and core. This analysis showed that the absence counts for TEs was higher than for an average 100 bp window, but there is no difference for the distribution of TE absence counts between those classified as LS versus core (Wilcoxon rank sum, p=0.99) (*Figure 6E,F*). Interestingly, this was not the case for genes, where the majority (64.4%) of core classified genes were present in all 42 strains, while only 43.1% of LS classified genes were present in all strains (Wilcoxon rank sum, p<2.2e-16; core mean absence count of 0.6 and LS mean of 10.3) (*Figure 6G,H*). These results suggest that TEs are generally variable between strains regardless of their genomic location, but the likelihood of a gene being absent varies significantly based on its location in a core or LS region. Analyzing genes based on functional categories showed that the number of in planta-induced genes in LS regions doubled from the old to new classification, and while these genes were overrepresented in the old designations ($X^2$ = 9.94, p=0.002), this overrepresentation was even more pronounced for the new LS regions ($X^2$ = 29.96, p<0.000001) (*Supplementary file 1*- table 11). As expected, we found that in planta induction for LS genes was significantly higher than for core genes (Mann-Whitney U-test, p-value=1.34e-50) (*Figure 6I*). Even though effectors were not overrepresented in the old LS classification ($X^2$ ~0, p=1), they were present in new LS regions far greater than by chance, increasing 3.5 times compared to the old classification ($X^2$ = 11.18, p=0.0008) (*Supplementary file 1*- table 12). The largest change was observed for genes coding proteins with secretion signals. Using the old classification, the LS regions were actually significantly underrepresented for genes coding secreted proteins ($X^2$ = 27.05, p<0.000001). This result could have a number of explanations, but under the assumption that LS regions contain a significant portion of genes involved in inter-species interactions, this analysis suggests that the old classification was missing important genes. Indeed, while the new LS classification approximately doubled the number of genes designated as LS (494 to 998), the number of LS genes with secretion signals increased 5-fold (20 to 109) ($X^2$ = 3.63, p=0.05) (*Supplementary file 1*- table 13).

Collectively, these analyses suggest that ML algorithms can be used to predict new LS regions based on epigenetic and physical DNA accessibility data. The identification of potentially new LS regions missed in the original classification provides new avenues to identify proteins important for host infection and adaptation. Our predictions were validated by showing enrichment for functional categories of genes known to be important for infection and host adaptation. Our results further show that the expanded classifications represent regions of the genome that are more variable across strains of the species and uncover the new finding that LS genes in particular experience greater PAV in *V. dahliae*. These results support that genome structure is influencing genome function and demonstrates a ML approach for predictive biology that advances our understanding of genome biology.

## Unsupervised genome clustering using chromatin data supports functional differences for core and LS classification

Using the described supervised learning approach, we were able to identify new regions of the genome as LS, and subsequently validated that these new regions fit the characteristics of LS function. To further confirm these results and define the functional genome, uniform manifold approximation and projection (UMAP) (*McInnes et al., 2018a*) was employed for dimensional reduction of TEs and genes based on transcriptional, chromatin and DNA openness data. The significance of this alternative approach is that it is unsupervised, and does not rely on, or is influenced by, prior LS and core classifications. Under the hypothesis that genome structure influences genome function, a prediction is that LS and core classification (evolutionary function) should show a non-random spatial association when layered on-top of the UMAP clustering (genome structure and function data). This approach generated three distinct UMAP groups for TEs, which we termed Group1, 2 and 3 (*Figure 7A*). When the LS and core classification were applied to the UMAP groups, Group1 and Group2 displayed significant non-random associations of core- and LS-designated elements respectively (x-axis p-value=1.77e-38 and y-axis p-value=9.0e-80, Mann-Whitney U test) (*Figure 7A*). Additionally, the core and LS elements were enriched in Group1 and Group2, respectively, ($X^2$ = 348.84, p=1.78e-76) (*Supplementary file 1*- table 14). To understand the UMAP groups, each genomic variable used for UMAP was summarized across the three groups (*Figure 7—figure supplement 1*). The TEs in Group1 have the lowest GC content, transcriptional activity, and DNA openness, along with the highest CRI, H3K9me3 signal, and DNA methylation (*Supplementary file 1*- table 15, Kruskal-Wallis test; *Figure 7—figure supplement 1*, Conover test with Holm correction). Based on these characteristics, we conclude that Group1 TE elements from UMAP are largely heterochromatic. The TE elements in Group2 and Group3 are more similar to one another based on the per-variable analysis, although many statistically significantly differences exist (*Figure 7—figure supplement 1*). These findings show that unsupervised genomic clustering on functional and structural data can recapitulate a large part of our previously defined core and LS regions. We interpret this data as supporting a link between genome organization on a physical level (i.e. epigenetics and DNA accessibility), genome function (i.e. transcriptional activity), and genome adaptation to the environment (i.e. LS and core regions). Interestingly, while there was no difference in PAV for TE elements classified as core and LS (*Figure 6E*), we did find significant PAV differences for the three UMAP groups (Kruskal-Wallis H statistic = 593.73, p-value=1.18e-129) (*Figure 7B*, Conover test with Holm correction). Within the three UMAP groups, there were also significant differences for PAV between LS and core elements (*Figure 7C*, Mann-Whitney U test and Holm correction). Specifically, the median absence count for all core TEs was 15 (*Figure 6G*), but the median count is less than one for the core TEs in UMAP Group2 and Group3, which makeup 39.4% of the core TEs (*Figure 7C*). It is not clear what accounts for the core TE elements in UMAP Group2 and Group3, which are less defined by heterochromatic characteristics but experience less absence variation across the analyzed *V. dahliae* strains.

We similarly analyzed genes using UMAP, under the same prediction that LS and core elements would show spatial grouping within the UMAP analysis. Here, UMAP grouped the majority of the genes in the genome (89.9%) into a single cluster (*Figure 7D*). Within this group, we observed significant local-clustering of LS classified genes (x-axis p-value=5.45e-221 and y-axis p-value=1.84e-28, Mann-Whitney U test and Holm correction), which can be seen as an 'island' in the upper left section of the group (*Figure 7D*). Based on this, the large group was further sub-divided based on a visible separation, and the UMAP results were analyzed based on the resulting three groups (*Figure 7E*). The UMAP Group1 genes contained four times more LS genes than expected ($X^2$ = 2119.4, p-value=0) (*Supplementary file 1*- table 16), and Group1 containing 75% of the LS genes. Furthermore, Group1 genes had the lowest GC content and transcription in axenic culture, and the highest H3K9 and H3K27 trimethylation (*Supplementary file 1*- table 17, Kruskal-Wallis test; *Figure 7—figure supplement 2*, Conover test with Holm correction). Interestingly, the Group1 genes also had the highest in planta gene induction, which was not a variable used for UMAP creation, further validating that UMAP generated groups with relevant biological differences (Kruskal-Wallis H statistic = 1582.09, p-value=0) (*Figure 7F*, Conover test with Holm correction). There was no significant difference for gene absence counts between the three groups (Kruskal-Wallis H statistic = 4.18, p-value=0.09) (*Figure 7—figure supplement 3A,B*), however this was influenced by the majority of genes having a

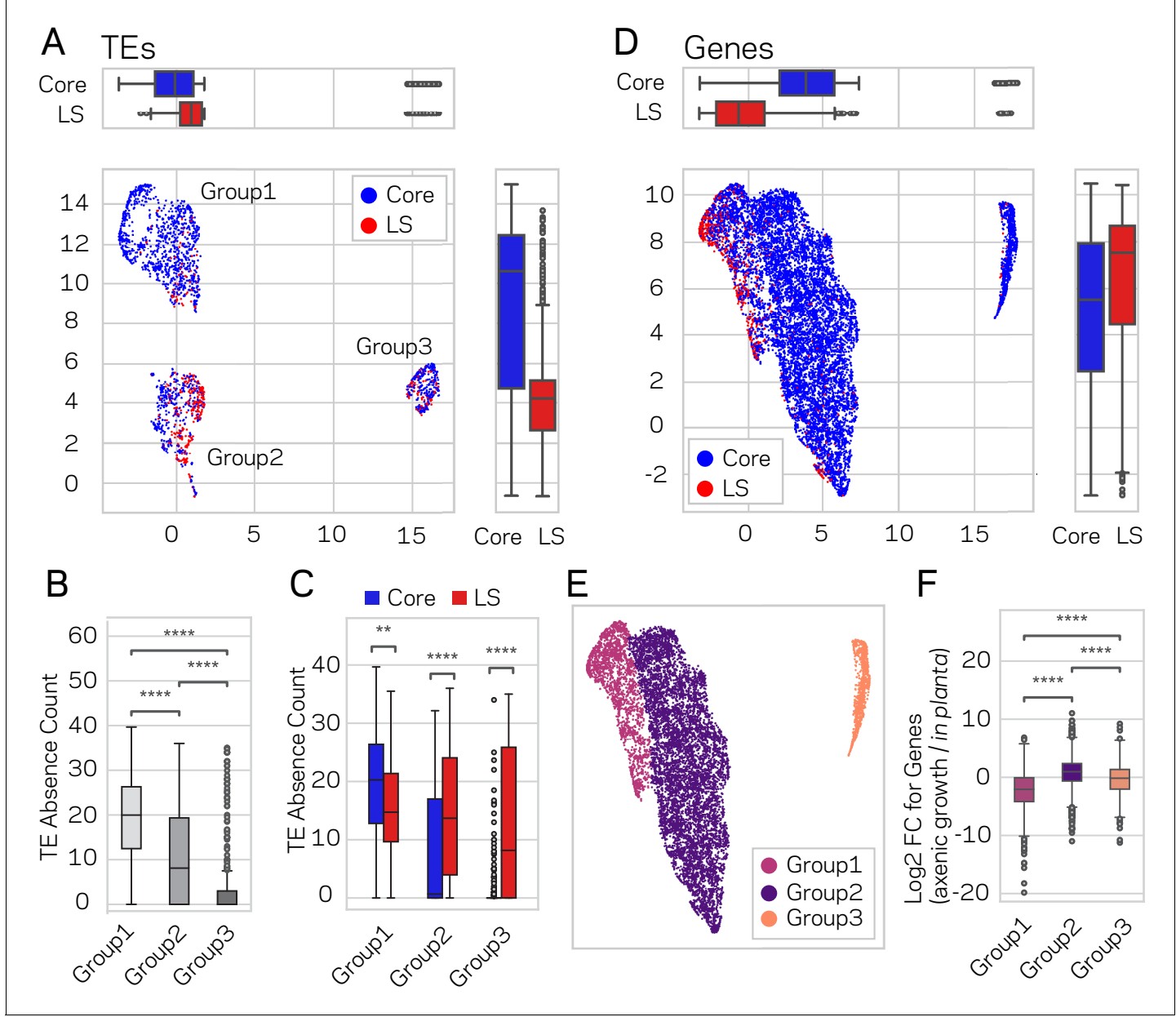

**Figure 7.** Genome-wide UMAP groups details that functional elements labeled core and LS have different epigenetic and DNA characteristics. (A) Uniform Manifold Approximation and Projection (UMAP) clustering of individual *V. dahliae* TEs, color coded for core (blue) and LS (red). UMAP clustering in two dimensions resulted in the identification Group1, 2, and 3 elements. Boxplots are shown opposite of the x- and y-axis to quantify the UMAP designation of the LS and core elements. Statistical difference measured using Mann-Whitney U test for UMAP labeling on the x-axis, p-value=1.77e-38, and y-axis, p-value=9.04e-80. (B) Boxplot for TE absence counts for UMAP Group1, 2, and 3 elements. Statistical difference measured using Conover's test and Holm multiple-test correction of p-values (C) Boxplot for TE absence counts for LS and core elements in UMAP Group1, 2, and 3. Statistical difference measured using Mann-Whitney U test and Holm multiple-test correction of p-values. (D) Similar UMAP clustering as shown in (A), but performed using genes as the clustering elements shown as individual dots. Marginal boxplots shown as in (A), x-axis p-value=5.45e-221, and y-axis p-value=1.84e-28. (E) UMAP gene clustering, color coded to show three groups. (F) Boxplot for in planta gene induction for UMAP Group1, 2, and 3 genes. Statistical difference measured using Conover's test and Holm multiple-test correction of p-values. **, p-value<0.01; ****, p-value<0.0001. The online version of this article includes the following source data and figure supplement(s) for figure 7:

**Source data 1.** Genomic data and UMAP group for TEs.
**Source data 2.** Genomic data and UMAP group for genes.
**Figure supplement 1.** Multiple comparisons of transposable elements (TEs) in UMAP groups for genomic variables.
**Figure supplement 2.** Multiple comparisons of Genes in UMAP groups for genomic variables.
**Figure supplement 3.** UMAP groupings vary significantly for Absence across V.

zero absence count (i.e. are conserved). When only genes with an absence count greater than 0 were analyzed (i.e. the gene was missing from at least one strain), Group1 genes had a significantly higher mean absence count (Group1 = 13.4; Group 2 = 1.9; Group 3 = 3.1, Kruskal-Wallis H statistic = 583.06, p-value=2.46e-127) (*Figure 7—figure supplement 3C,D*, Conover test with Holm correction). Thus, the UMAP Group1 genes have characteristics of heterochromatin when grown axenically, are enriched for LS classified genes, display significantly higher in planta gene induction, and the genes have the highest absence counts across the analyzed *V. dahliae* strains. These results show that unsupervised UMAP groupings are able to capture functionally relevant biological groupings based on genomic and chromatin data, a previously unreported link.

## Discussion

Significant efforts to detail genomes of filamentous pathogens, to understand variation within species, and to a lesser extent to examine epigenetic modifications, have increased our understanding of genome function in this important group of organisms (*Shi-Kunne et al., 2018*; *Chen et al., 2018*; *Mondo et al., 2017*). This is required to broaden our understanding of eukaryotic genome function and in order to combat emerging pathogens. Here, we present a detailed analysis of the epigenome and physical DNA accessibility of the vascular wilt pathogen *V. dahliae* and link these analyses to previous characterizations of genomic regions contributing to host colonization and adaptation (*Klosterman et al., 2011*; *de Jonge et al., 2013*; *Faino et al., 2016*; *Shi-Kunne et al., 2018*). A clear picture emerges in which the core genome is organized into heterochromatic and euchromatic regions. The heterochromatin is characterized by a high density of TEs with low GC content, high levels of DNA and H3K9 methylation, low DNA accessibility and clear signatures of RIP mutations at repetitive sequences. The euchromatin regions are opposite in all characteristics, and this collective description is consistent with previous research in many other eukaryotic genomes (*Sexton et al., 2012*; *Bannister et al., 2001*; *Cam et al., 2005*). Interestingly, we provide evidence that previously defined LS regions of the genome, characterized for their role in contributing to host infection, exist in a unique chromatin state. LS regions have higher TE density than the euchromatic regions, yet are devoid of DNA and H3K9 methylation. Furthermore, LS regions have higher DNA accessibility than the core heterochromatic regions and are more transcriptionally active, but they are less accessible than the 'true' euchromatic gene-rich core regions. Notably, LS regions are characterized as having a strong association with H3K27me3, similar to the discovery that SM gene clusters are enriched at H3K27me3 regions in *F. graminearum* (*Connolly et al., 2013*). Our results demonstrate that LS regions are by definition not heterochromatic, as they are far more accessible than the true heterochromatin, and yet they typically contain many heterochromatin features. These observations are akin to previous descriptions of facultative heterochromatin (*Huisinga et al., 2006*), but to our knowledge few studies have linked unique chromatin states to adaptive genomic regions. We believe this observation presents new hypotheses into the occurrence, formation and maintenance of adaptive regions of the genome. How chromatin interacts with evolutionary forces to shape organism fitness is an important question in the pursuit of understanding the genome.

Recent work from our team has shown that the eight centromeres in *V. dahliae* are significantly enriched in H3K9me3 and DNA methylation and are devoid of protein coding genes (*Seidl et al., 2020*). These regions, which are critically important for genome function, appear as true heterochromatin under the conditions tested here. Our analysis of the protein HP1, an important protein for heterochromatin formation, demonstrates it is required for DNA methylation in *V. dahliae*, consistent with reports in other fungi (*Freitag et al., 2004*). The loss of DNA methylation in Δ*hp1* is likely driven by the need for HP1 to bind H3K9me3 and direct DNA methylation. An interesting observation here is that only some of the TEs in the genome have H3K9me3, DNA methylation and signatures of RIP. The process of RIP has been shown to occur during meiosis, yet *V. dahliae* is presumed to reproduce asexually. One explanation for this is that the ancestral status was characterized by sexual reproduction and included RIP, and TEs present at that time remain in the genome as heterochromatin, some of which at centromeres. Under this explanation, the occurrence of TEs in the genome which are not marked by H3K9me3, DNA methylation, and do not appear affected by RIP, would have arisen in the genome after sexual reproduction was lost. This would be consistent with the occurrence of relatively newer TEs in *V. dahliae* residing in LS regions and contributing to adaptive evolution (*Faino et al., 2016*). However, other possibilities cannot be ruled out, such that sexual reproduction

does occur in *V. dahliae*, albeit incidentally in local field populations that underly the many asexual lineages that are presently mostly observed. It is also possible that RIP occurs outside of meiosis, for example during parasexual exchange while in a diploid state (*Forche et al., 2008*). Further experimental research is needed to further understand these mechanistic possibilities.

Our results support the hypothesis that chromatin structure underlies genome function. This is clearly true for the functional centromere, in which centrochromatin is defined by epigenetics, diverse DNA sequence, and dynamic configuration in the cell (*Friedman and Freitag, 2017*; *Nagpal and Fierz, 2020*). Another example supporting this hypothesis exists in *Z. tritici*, where accessory chromosomes are enriched for specific histone modifications, but it remains to be determined how these accessory chromosomes impact genome function (*Schotanus et al., 2015*). The results presented here suggest that chromatin modifications and DNA accessibility may influence genome evolution, not only via transcriptional control, but also by contributing to genome variation. Based on the observed association between chromatin and function, we were able to predict LS regions using machine learning. The ML algorithms, trained on H3K9 and H3K27 methylation, RNA-sequencing, TE density and DNA accessibility data, show how these descriptions of chromatin can be used to classify DNA segments as core versus LS with high recall (i.e. sensitivity). Our results indicated low precision for ML classification, owing to the relatively high number of false positives. It was not our original intention to try and identify new LS regions using ML, and thus our approach was not necessarily optimized to identify novel LS regions. Despite this, we were able to identify and validate new LS regions based on enrichment for in planta-induced genes, presumed effectors, genes coding secreted peptides and genes with higher PAV, thereby improving the classification and nearly doubling the amount of LS DNA. It is a remarkable finding that, through the use of machine learning, we could substantially extend our knowledge of the *V. dahliae* genome and identify as of yet unconsidered genomic regions and genes, which likely contribute to adaptive traits. In addition to the supervised binary classification, we employed unsupervised uniform manifold approximation and projection (UMAP) to cluster the genome, without any explicit information pertaining to core or LS regions. The UMAP approach reduced our multidimensional representation of the genomic data into a simple two-dimension scatterplot. Analyzing the UMAP groups for both TEs and genes showed that specific groups were enriched for LS elements. This is remarkable given that UMAP is an independent analysis from the ML approach, and was not built using a priori LS or core designation. As such, this provides strong evidence that genomic and chromatin features have a strong association with DNA conferring specific biological function and evolutionary relevance. We note that the original definition of LS (*Klosterman et al., 2011*; *de Jonge et al., 2013*) is now a too limited description of the genomic regions described here. Our hypothesis is that these regions make a substantial contribution to host adaptation because they contain many known effectors and genes induced during in planta growth, code for many proteins with secretion signals, and they show significantly more PAV between the analyzed strains. As such, we now propose to refer to these regions as 'adaptive genomic regions'.

It is currently not possible to extend our machine learning predictions to additional filamentous pathogen genomes, as the necessary data are not currently publicly available. However, for many filamentous plant pathogens it is clear that genome variation on multiple scales, from SNPs to large structural variation, are not uniformly distributed in the genome (*Connolly et al., 2013*). Recent reports from the fungal pathogen *Z. tritici* addressed the role of genome stability and H3K27me3 during asexual reproduction (*Möller et al., 2019*; *Möller et al., 2018*). During experimental evolution, individual strains of *Z. tritici* readily lose accessory chromosomes. The authors observed that a mutant lacking the enzyme responsible for H3K27me3 showed less accessory chromosome loss and concluded that H3K27me3 destabilizes chromosome structure (*Möller et al., 2019*). However, accessory chromosome losses were clearly biased in their individual frequency and changes were not reported for core chromosomes, despite H3K27me3 being found at high levels on accessory and regions of core chromosomes (*Schotanus et al., 2015*). Therefore, the observed genome destabilization requires additional determinants in conjunction with H3K27me3 which remain to be discovered. Results presented here suggest that DNA and histone methylation marks, along with physical DNA accessibility are important additional determinants to distinguish functionally, and potentially evolutionarily relevant regions of the genome.

We acknowledge that our model does not strictly differentiate all regions of the *V. dahliae* genome, as there were LS and core regions that had similar overall chromatin profiles, and therefore

these features alone are not sufficient. It is possible that variability in growth conditions or tissue used for genomic data collection affected our results. Our results also relied on the original LS classification for model training, which given the lack of ground truth, may have introduced error. Future efforts to develop higher confidence training data may improve this approach. Another explanation is that the formation of adaptive genomic regions is not fully deterministic. Evolution is a stochastic process, and it seems unlikely that adaptive genomic regions will be formed in absolute terms. Rather, it is more likely to be a probabilistic process, in which specific chromatin and physical status increases the likelihood for formation and maintenance of adaptive regions. Along these lines, it is important to note that while binary classification is useful, this model is likely overly simplified as genomic regions probably exists on a continuum for adaptive potential. Further experimental evidence is needed to to fully understand if different chromatin states directly impact genome evolution. Data present here support this hypothesis, and offer an exciting new link between the epigenome, physical DNA accessibility and adaptive genome evolution.

# Materials and methods

## Key resources table

| Reagent type (species) or resource | Designation | Source or reference | Identifiers | Additional information |
|---|---|---|---|---|
| Strain, strain background (*V. dahliae*) | JR2, wild-type | PMID:26286689 | | Fungal Biodiversity Center (CBS), 143773 |
| Strain, strain background (*V. dahliae*) | JR2, Δ*hp1* | https://doi.org/10.1101/2020.08.26.268789 | | |
| Antibody | Rabbit anti H3K9me3 (Polyclonal) | Active Motif | 39765 | ChIP (1:200) |
| Antibody | Rabbit anti H3K27me3 (Polyclonal) | Active Motif | 39155 | ChIP (1:200) |
| Recombinant DNA reagent | pRF-HU2 | *Frandsen et al., 2008* | | USER-cloning |
| Commercial assay or kit | EZ DNA Methylation-Gold kit | Zymo Research | D5005 | |
| Commercial assay or kit | Nextera DNA library Preparation | Illumina | FA-121–1030 | |

## Fungal growth and strain construction

*V. dahliae* strain JR2 (CBS 143773) was used for experimental analysis (*Faino et al., 2015*). The strain was maintained on potato dextrose agar (PDA) (Oxoid, Thermo Scientific, CM0139) and grown at 22°C in the dark. For liquid grown cultures, conidiospores were collected from PDA plates after approximately two weeks and inoculated into flasks containing the desired media at a concentration of $2 \times 10^4$ spores per mL. Media used in this study include PDA, half-strength Murashige and Skoog plus vitamins (HMS) (Duchefa-Biochemie, Haarlem, The Netherlands) medium supplemented with 3% sucrose and xylem sap (abbreviated, X) collected from greenhouse grown tomato plants of the cultivar Moneymaker. Liquid cultures were grown for four days in the dark at 22°C and 160 RPM. The cultures were strained through miracloth (22 µm) (EMD Millipore, Darmstadt, Germany), pressed to remove liquid, flash frozen in liquid nitrogen and ground to powder with a mortar and pestle. Samples were stored at −80°C if required prior to nucleic acid extraction.

The Δ*hp1* strain was constructed as previously described (*Santhanam, 2012*). Briefly, the genomic DNA regions flanking the 5′ and 3′ HP1 coding sequence were amplified (*left border*, For. Primer, 5′-GGTCTTAAUGACCTGAAGAATCGAGCAAGGA and Rev. primer, 5′-GGCATTAAUATGAAAG-CACCGGGATTTTTCT; *right border*, For. Primer, 5′-GGACTTAAUATGCTGTTGGGAGGCAGAA TAA Rev. primer, 5′-GGGTTTAAUCCACGTAGATGGAGGGGTAGA). The PCR products were cloned in to the pRF-HU2 vector system (*Frandsen et al., 2008*) using USER enzyme following manufactured protocol (New England Biolabs, MA, USA). Correctly ligated vector was transformed into *Agrobacterium tumefaciens* strain AGL1 used for *V. dahliae* spore transformation (*Santhanam, 2012*). Colonies of *V. dahliae* growing on hygromycin B selection after 5 days were moved to individual plates containing PDA and hygromycin B. Putative transformants were screened using

PCR to check for deletion of the HP1 sequence (For. Primer, 5'- AATCCCGCAAGGGAAAAGAGAC and Rev. primer, 5'- CGTGTGCTTTGTCTTCTGACCA) and the integration of the hygromycin B sequence (For. Primer, 5'- TGGAATATGCCACCAGCAGTAG and Rev. primer, 5'- GGAGTCGCA TAAGGGAGAGCG) at the specific locus.

## Bisulfite sequencing and analysis

The wild-type *V. dahliae* strain and Δ*hp1* were grown in liquid PDA for three days, flash frozen and collected as described earlier. DNA was extracted from a single sample of wild-type strain JR2 and Δ*hp1* and sent to the Beijing Genome Institute (BGI) for bisulfite conversion, library construction and Illumina sequencing. Briefly, the DNA was sonicated to a fragment range of 100–300 bp, end-repaired and methylated sequencing adapters were ligated to 3' ends. The EZ DNA Methylation-Gold kit (Zymo Research, CA, USA) was followed according to manufacturer guidelines for bisulfite conversion of non-methylated DNA. Libraries were paired-end 100 bp sequenced on an Illumina HiSeq 2000.

Whole-genome bisulfite sequencing reads were analyzed using the BSMAP pipeline (v. 2.73) and methratio script (*Xi and Li, 2009*). The results were partitioned into CG, CHG and CHH cytosine sites for analysis. Only cytosine positions containing greater than four sequencing reads were included for analysis. Methylation levels were summarized as weighted methylation percentage, calculated as the number of reads supporting methylation over the number of cytosines sequenced or as fractional methylation, calculated as the number of methylated cytosines divided by all cytosine positions (*Schultz et al., 2012*). For fractional methylation, a cytosine was considered methylated if it was at least 5% methylated from all the reads covering that cytosine. As such, weighted methylation captures quantitative aspects of methylation, while fractional methylation is more qualitative. Weighted and fractional methylation were calculated over intervals described in the text, including genes, promoters (defined as the 300 bp upstream of the translation start site), TEs and 10 kb windows. For each feature, weighted and fractional methylation were calculated from the sum of the mapped reads or the sum of the positions, respectively, over the analyzed region. Two sample comparisons were computed using base R (*R Development Core Team, 2019*) to compute the non-parametric Mann-Whitney U test (equivalent to the two-sample Wilcoxon rank-sum test) and Holm multiple testing correction was used for the associated p-values. Principle component analyses were computed in R using the packages FactoMineR (v 1.42) (*Le et al., 2008*) and factoextra (v 1.0.5) (*Kassambara and Mundt, 2017*).

## Transposable element annotation

Repetitive elements were identified in the *V. dahliae* stains JR2 genome assembly (*Faino et al., 2015*) as well as in three other high-quality *V. dahliae* genome assemblies (*Shi-Kunne et al., 2018*) using a combination of LTRharvest (*Ellinghaus et al., 2008*) and LTRdigest *Steinbiss et al., 2009* followed by de novo identification of RepeatModeler (*Smit and Hubley, 2015*). Briefly, LTR sequences were identified (recent and ancient LTR insertions) and subsequently filtered, for example for occurrence of primer binding sites or for nested insertions (see procedure outlined by Campbell and colleagues for details [*Campbell et al., 2014*]). Prior to the de novo prediction with RepeatModeler, genome-wide occurrences of the identified LTR elements are masked. Predicted LTR elements and the de novo predictions from RepeatModeler were subsequently combined, and the identified repeat sequences of the four *V. dahliae* strains were clustered using vsearch (80% sequence identity, search on both strands; v 2.4.4) (*Rognes et al., 2016*). A non-redundant *V. dahliae* repeat library that contained consensus sequences for each cluster (i.e. repeat family) was constructed by performing multiple sequence alignments using MAFFT (v7.271) *Katoh and Standley, 2013* followed by the construction of a consensus sequence as described by *Faino et al., 2016*. The consensus repeat library was subsequently manually curated and annotated (Wicker classification [*Wicker et al., 2007*]) using PASTEC (default databases and settings; search in the reverse-complement sequence enabled) (*Hoede et al., 2014*), which is part of the REPET pipeline (v2.2) (*Flutre et al., 2011*), and similarity to previously identified repetitive elements in *V. dahliae* (*Faino et al., 2015*; *Amyotte et al., 2012*). The occurrence and location of repeats in the genome assembly of *V. dahliae* strain JR2 were determined using RepeatMasker (v 4.0.7; sensitive option). The RepeatMasker output was post-processed using 'One code to find them all' (*Bailly-Bechet et al., 2014*) which supports

the identification and combination of multiple matches (for instance due to deletions or insertions) into combined, representative repeat occurrences. We only further considered matches to the repeat consensus library, and thereby excluded simple repeats and low-complexity regions. To estimate divergence time of TEs, each individual copy of a TE was aligned to the consensus of its family using needle, which is part of the EMBOSS package (*Rice et al., 2000*). Sequence divergence between the TEs and the TE-family consensus was corrected using the Jukes-Cantor distance, with a correction term that accounts for insertions and deletions (*Jukes, 1969*; *Van de Peer et al., 1990*). The composite RIP index (CRI) was calculated as previously described (*Lewis et al., 2009*). Briefly, CRI was determined by subtracting the RIP substrate from the RIP product index, which are defined by dinucleotide frequencies as follows: RIP product index = (TpA/ApT) and the RIP substrate index = (CpA + TpG/ApC + GpT). Positive CRI values indicate the analyzed sequences were subjected to the RIP process. For TE analysis, elements that are less than 100 bp were removed.

## RNA-sequencing and analysis

*V. dahliae* strain JR2 (CBS 143773) was grown in triplicate liquid media PDB, HMS, and xylem sap as described. RNA extraction was carried out using TRIzol (Thermo Fisher Science, Waltham, MA, USA) following manufacturer guidelines. Following RNA re-suspension, contaminating DNA was removed using the TURBO DNA-free kit (Ambion, Thermo Fisher Science, Waltham, MA, USA) and RNA integrity was estimated by separating 2 µL of each sample on a 2% agarose gel and quantified using a Nanodrop (Thermo Fisher Science, Waltham, MA, USA) and stored at −80˚C. Library preparation and sequencing was carried out at BGI. Briefly, mRNA were enriched based on polyadenylation purification and random hexamers were used for cDNA synthesis. RNA-sequencing libraries were constructed following end-repair and adapter ligation protocols and PCR amplified. Purified DNA fragments were single-end 50 bp sequenced on an Illumina HiSeq 2000.

Reads were mapped to the *V. dahliae* stain JR2 genome assembly (*Faino et al., 2015*) using STAR (v 2.6.0) with settings (`–sjdbGTFfeatureExon exon`, `–sjdbGTFtagExonParentTranscript Parent`, `–alignIntronMax 400`, `–outFilterMismatchNmax 5`, `–outFilterIntronMotifs RemoveNoncanonical`) (*Dobin et al., 2013*). Mapped reads were quantified using the `summarizeOverlaps` (*Lawrence et al., 2013*) and variance stabilizing transformation (*vst*) features of DESeq2 (*Love et al., 2014*). For TE analysis, the coordinates of the annotated TEs were used as features for read counting. To perform RNAseq analysis over whole genome 10 kb regions, raw mapped reads were summed over 10 kb bins using bedtools (v 2.27) (*Quinlan and Hall, 2010*) and converted to Transcripts Per Million (TPM) (*Wagner et al., 2012*) and averaged over the three reps for analysis.

## Chromatin immunoprecipitation and sequencing and analysis

*V. dahliae* strain JR2 was grown in PDB and samples collected as described. Approximately 400 mg ground material was resuspended in 4 ml ChIP Lysis buffer (50 mM HEPES-KOH pH7.5, 140 mM NaCl, 1 mM EDTA, 1% Triton X-100, 0.1% NaDOC) and dounced 40 times in a 10 cm$^3$ glass tube with tight fitting pestle on 800 power with a RZR50 homogenizer (Heidolph, Schwabach, Germany), followed by five rounds of 20 s sonication on ice with 40 s rest between rounds with a Soniprep 150 (MSE, London, UK). Samples were redistributed to 2 ml tubes and pelleted for 2 min at max speed in tabletop centrifuge. The supernatants were combined, together with 20 µl 1M CaCl2 and 2.5 µl MNase, and after 10 min of incubation in a 37˚C water bath with regular manual shaking, 80 µl 0.5M EGTA was added and tubes were put on ice. Samples were pre-cleared by adding 40 µl Protein A Magnetic Beads (New England Biolabs, MA, United States) and rotating at 4˚C for 60 min, after which the beads were captured, 1 ml fractions of supernatant were moved to new 2 ml tubes containing 5 µl H3K9me3 or H3K27me3 antibody (ActiveMotif; #39765 and #39155) respectively and incubated overnight with continuous rotation at 4˚C. Subsequently, 20 µl protein-A magnetic beads were added and incubated for 3 hr at 4˚C, after which the beads were captured on a magnetic stand and subsequently washed with 1 ml wash buffer (50 mM Tris HCl pH 8, 1 mM EDTA, 1% Triton X-100, 100 mM NaCl), high-salt wash buffer (50 mM Tris HCl pH 8, 1 mM EDTA, 1% Triton X-100, 350 mM NaCl), LiCl wash buffer (10 mM Tris HCl pH8, 1 mM EDTA, 0.5% Triton X-100, 250 mM LiCl), TE buffer (10 mM Tris HCl pH 8, 1 mM EDTA). Nucleosomes were eluted twice from beads by addition of 100 µl pre-heated TES buffer (100 mM Tris HCl pH 8, 1% SDS, 10 mM EDTA, 50 mM

NaCl) and 10 min incubation at 65°C. 10 mg /ml 2 µl Proteinase K (10 mg /ml) was added and incubated at 65°C for 3 hr, followed by chloroform clean-up. DNA was precipitated by addition of 2 volumes 100% ethanol, 1/10th volume 3 M NaOAc pH 5.2 and 1/200th volume 120 mg/ml glycogen, and incubated overnight at −20°C. Two ChIP replicates were performed for each antibody from independently grown samples. Sequencing libraries were prepared using the TruSeq ChIP Library Preparation Kit (Illumina) according to instructions, but without gel purification and with use of the Velocity DNA Polymerase (BioLine, Luckenwalde, Germany) for 25 cycles of amplification. Single-end 125 bp sequencing was performed on the Illumina HiSeq2500 platform at KeyGene N.V. (Wageningen, the Netherlands).

Reads were mapped to the reference JR2 genome, using BWA-mem with default settings (*Li, 2013*). For ChIP and ATAC-seq mapping, three regions of the genome were masked due to aberrant mapping, possibly owing to sequence similarity to the mitochondrial genome (chr1:1–45000, chr2:3466000–3475000, chr3:1–4200). This is similar to what is described as blacklisted regions in other eukaryotic genomes (*Amemiya et al., 2019*). The raw mapped reads were counted either over the TE coordinates or 10 kb intervals for the two separate analyses. The raw mapped reads were converted to TPM and the average of the two ChIP-seq replicates were used for analysis.

## Assay for transposase-accessible chromatin (ATAC)-sequencing and analysis

The *V. dahliae* strain JR2 was grown in PDB liquid media as described. Mycelium was collected, filtered, rinsed and flash frozen in liquid nitrogen. The ATAC-seq procedure was carried out mainly as described previously (*Buenrostro et al., 2015*). Nuclei were collected by resuspending ground mycelium in 5 mL of ice-cold Nuclei Isolation Buffer (NIB) (100 mM NaCl, 4 mM NaHSO$_4$, 25 mM Tris-HCl, 10 mM MgSO$_4$, 0.5 mM EDTA, 0.5% NP-40 including protease inhibitors added at time of extraction, 2 mM Phenylmethanesulfonyl fluoride (PMSF), 100 µM Leupeptin, 1 µg/mL Pepstatin, 10 µM E-64). The homogenate was layered onto 10 mL of an ice-cold sucrose-Ficoll gradient (bottom layer 5 mL of 2.5M sucrose in 25 mM Tris-HCl, 5 mL 40% Ficoll 400 (GE Biosciences Corporation, NJ, USA)). Nuclei were separated into the lower phase by centrifugation at 2000 g for 30 min at 4°C. The upper layer was discarded and the lower phase (~4 mL) moved to another collection tube containing 5 mL of ice-cold NIB. Nuclei were pelleted at 9000 g for 15 min at 4°C and re-suspended in 3 mL of NIB. The integrity of the nuclei and their concentration in the solution were estimated by DAPI staining (DAPI Dilactate 5 mg/mL, used at a 1/2000 dilution for visualization) and counted on a hemocytometer. A total of 200,000 nuclei were transferred to a 1.5 mL microfuge tube, and nuclei pelleted at 13,000 g for 15 min at 4°C and resuspended in the transposition reaction (20 µL of 2x Nextera reaction buffer, 0.5 µL of Nextera Tn5 Transposase, 19.5 µL of nuclease-free H$_2$0) (Illumina, Nextera DNA library Preparation kit FA-121–1030) and the reaction was carried out for 5 min at 37°C. Empirical testing showed this Tn5 incubation period and nuclei density resulted in optimal DNA fragmentation, and a single sample was used for further library preparation and sequencing. The reaction was halted, and fragmented DNA purified using a MinElute PCR purification kit (Qiagen, MD, USA). The eluted DNA was amplified in reaction buffer (10 µL of transposased DNA, 10 µL nuclease-free H$_2$0, 2.5 µL forward PCR primer (5'-AATGATACGGCGACCACCGAGATCTACACTCG TCGGCAGCGTCAGATGTG), 2.5 µL reverse PCR primer (CAAGCAGAAGACGGCATACGAGATTTC TGCCTGTCTCGTGGGCTCGGAGATGT) and 25 µL NEBnext High-Fidelity 2x PCR Master Mix (New England Biolabs, MA, United States)) using thermo-cycler conditions described in *Buenrostro et al., 2015* for a total of nine cycles. Amplified library was purified using the MinElute PCR Purification Kit (Qiagen, MD, USA) and paired-end 100 bp sequenced on an Illumina HiSeq4000.

Reads were mapped to the reference JR2 genome with the described blacklisted regions masked, using BWA-mem with default settings (*Li, 2013*). The mapped reads were further processed to remove duplicates reads arising from library prep and sequencing using Picard toolkit *markDuplicates* (*Picard Toolkit, 2018*). The mapped reads were counted either over the TE coordinates or 10 kb intervals for the two separate analyses using bedtools *multicov* (v 2.27) (*Quinlan and Hall, 2010*). The reads were converted to TPM values and those numbers used for analysis.

## Machine learning and assessment

The machine learning algorithms were implemented using the classification and regression training (caret) package in R (*R Development Core Team, 2019*; *Kuhn, 2008*). The full set of genomic data was used to create a data frame comprising the genome in 10 kb segments as rows and the individual collected variables as columns. The regions were classified as core or LS based on the previous observations (*Faino et al., 2016*). The model for all algorithms was; classification = ATAC seq$_{TPM}$ + ChIP-H3K27me3$_{TPM}$ + ChIP-H3K9me3$_{TPM}$ + TE$_{density}$ + PDB-RNAseq$_{TPM}$. The data were split into 80% training (i.e. learning) and 20% testing (i.e. prediction), and the proportion of core and LS regions were kept approximately equal in the two splits. Only the training data was used for parameter tuning and testing. Four algorithms used tested- logistic regression (LR), random forest (RF), stochastic gradient boosting (GBM), and boosted classification tree (BCT). The models were assessed by testing various tuning parameters, which are model specific by performing three-time 10-fold cross-validation with the training data. Final parameter settings were determined based on the highest accuracy. The logistic regression model was implemented using method *glm* and family *binomial* and this method does not have parameters to tune. The random forest model was implemented using method *rf* and the tuning parameter 'Randomly Selected Predictors' (*mtry*) was tried at values 1, 2, 3, and 4, with a value of 1 giving the highest mean accuracy (supp fig ML#1). The stochastic gradient boost model was implemented using method *gbm* and the tuning parameters 'number of iterations' (*n.trees*) was tested at 50, 500, 1000; 'complexity of trees' (*interaction.depth*) was tested at 1, 5, 10; 'learning rate' (*shrinkage*) was tested at 0.01 and 0.001; and 'minimum node splitting' (*n.minobsinnode*) was tested at 1, and 5 (supp fig ML#2). The final stochastic gradient boost model used parameters *n.trees* = 500, *interaction.depth* = 5, *shrinkage* = 0.01, and *n.minobsinnode* = 5. The boosted classification tree model was implemented using method *ada* and the tuning parameters 'number of trees' (*iter*) was tested at 100, 1000, 3000; 'maximum tree depth' (*maxdepth*) was tested at 1, 5, 20; and the learning rate (*nu*) was held constant at 0.01 (supp fig ML#3). The final boosted classification tree model used parameters *iter* = 3000, *maxdepth* = 20, and *nu* = 0.01.

Once the best parameters were selected for each model, LS and Core classification was predicted on the unseen test data (i.e. 20% of original data never seen by the model). For binary classification, each data point is given a probability that it is of class positive or negative. A receiver operating characteristic (ROC) curve was generated using PRROC (*Grau et al., 2015*) and pROC (*Robin et al., 2011*) and used to determine the 'best' probability threshold at which data would be classified positive (i.e. LS). Based on the ROC, we used a threshold of 0.05 for the LR model, 0.08 for the RF model, 0.05 for the GBM model, and 0.00001 for the BCT model. Using these final model parameters and threshold for classification, final predictions were made for the test data presented in a confusion matrix (*Table 2*) and assessed using standard metrics for data retrieval (*Table 3*).

To saturate the genome in predictions, a total of 15 new training test splits (80:20) were generated, again maintaining the genome-wide proportion of core and LS regions in data set. The random forest and stochastic gradient boosting classifiers were used, based on their highest performance from the previous tests. The random forest model was implemented using method *rf* and parameter *mtry* = 3. The stochastic gradient boost model was implemented using method *gbm* and parameters *n.trees* = 500, *interaction.depth* = 5, *shrinkage* = 0.01, and *n.minobsinnode* = 5. For each round of training/testing, the trained model was used to predict LS and core classification using the test data with a probability threshold of 0.10. The predictions for each of the 15 runs were assessed using precision, recall and MCC metrics. A total of 124 regions did not receive a prediction because of this approach.

For each genomic region, a consensus designation was assigned based on the highest occurrence of core versus LS prediction across the 15 trials. If a similar number of core and LS predictions were made for a region across the 15 trials, the regions were designated as core to be conservative. This was done independently between the two models. Individual 10 kb consensus predictions for the GBM and RF models were assessed based on the original LS classification from *Faino et al., 2016* found in supplemental table S2. A final high confidence LS consensus was determined based on the results of the two models. Genomic regions that were predicted LS by both models were considered LS, while the rest were classified as core. Adjacent LS regions were concatenated, along with a conservative joining approach where a single core region was classified LS if it were flanked by two LS regions. The joining added 41 genomic regions (410 kb) to the LS genome.

For assessment of old and new LS designations, the following three categories of coding sequence were assessed; in planta induced genes, putative effectors and coding sequences for secreted peptides. The in planta induced genes were determined by mapping RNA sequencing reads from *V. dahliae* colonizing *Arabidopsis thaliana* at 21 days post-inoculation conducted in triplicate. Gene transcription levels in planta were compared to RNA-seq from in vitro cultivation in PDB using Kallisto quant with settings $-single$ -l 50 s 0.001 $-pseudobam$ (*Bray et al., 2016*). Differential gene expression between *A. thaliana* infection and PDB growth were determined using the DESeq2 package (*Love et al., 2014*), and genes up regulated in *Arabidopsis* compared to media with an adjusted p-value<0.05 were designated as in planta induced. Secreted peptides were predicted from the amino acid sequences of all annotated genes with SignalP (v5.0) (*Almagro Armenteros et al., 2019*). Putative effectors were predicted by further analyzing the amino acid sequences of secreted peptides using EffectorP (v2.0) (*Sperschneider et al., 2018*). For each functional category, a 2 × 2 contingency table was created for the number of genes in the functional category by the LS or core location for both the old and new LS classification. Pearson's chi-squared test and Yate's continuity correction were used to determine if the observed values were significantly different than expected. Yate's error correction reduces the chi-square value and is therefore conservative and less prone to false significance. The chi-square analysis and expected values were calculated using base R *chisq.test* (*R Development Core Team, 2019*).

## Analysis of presence absence variation

The LS and core designations were assessed for PAV across a collection of 42 *V. dahliae* strains (*Supplementary file 1*- table 18). PAV were identified using whole-genome alignments of DNA sequence reads from the 42 *V. dahliae* strains to the reference genome assembly of *V. dahliae* strain JR2 (*Faino et al., 2015*) using BWA-mem with default settings (*Li and Durbin, 2009*). Library artifacts were marked and removed using Picard Tools with -*MarkDuplicates* followed by -*SortSam* to sort the reads (*Picard Toolkit, 2018*). Raw read coverage was averaged per 100 bp non-overlapping window using the BEDtools -*multicov* function (*Quinlan and Hall, 2010*). To estimate presence or absence of a window per strain, we transformed the raw read coverage value to a binary classifier where a region with >= 10 reads indicate presence (1) and <10 reads indicate absence (0). For each window, the number of strains that were classified as absent were summed to get the 'absent count' value, which is easily interpretable as the number of strains for which the window was absent. To estimate absence counts for TEs and genes, the 100 bp absence count windows were intersected with TEs and genes using BEDtools -*intersect* where > 50% of the 100 bp window had to overlap with the feature (*Quinlan and Hall, 2010*). From these, a mean absence count was calculated per TE and gene and used for further analysis.

## Uniform manifold approximation and projection (UMAP) analysis

The UMAP algorithm for dimensional reduction (*McInnes et al., 2018a*) was implemented in Python3 (*McInnes et al., 2018b*; *Virtanen et al., 2020*). For TE analysis, the following variables were included when running UMAP: Jukes cantor, fraction of GC content, CRI, variance stabilized transformed RNA-seq from ½ MS grown fungal culture, log2 transformed TPM ChIP-seq for H3K9me3 and H3K27me3, log2 transformed TPM for ATAC-seq and log2 transformed weighted CG DNA methylation with the following parameters *n_neighbor* = 50, *n_components* = 2, *min_dist* = 0.1 and a *random_state* = 42. For gene analysis, the following variables were included when running UMAP: fraction of GC content, log2 transformed RNA-seq from PDB grown fungal culture, log2 transformed TPM ChIP-seq for H3K9me3 and H3K27me3, log2 transformed TPM for ATAC-seq and log2 transformed weighted CG DNA methylation with the following parameters *n_neighbor* = 100, *n_components* = 2, *min_dist* = 0.1 and a *random_state* = 42. Additional values for *n_neighbor* were checked to balance local versus global clustering. For genes, Group1 and Group2 were split based on visual assessment of the larger cluster, attempting to separate genes along what appeared as local clustering. The resulting two-dimensional values from UMAP *fit.transform* were used for plotting (*Hunter, 2007*; *Waskom et al., 2017*) and further statistical analysis (*van der Walt et al., 2011*; *McKinney, 2010*; *Pedregosa et al., 2011*). Pairwise post hoc tests were computed using either Mann-Whitney U-Test or Kruskal-Wallis test followed by post-hoc analysis with Conover's test using

Holm's multiple-testing correction, implemented from *scikit-posthocs* as *posthoc_mannwhitney* and *posthoc_conover* (*Terpilowski, 2019*).

## Acknowledgements

This work was supported in part by a European Molecular Biology Organization postdoctoral fellowship (EMBO, ALTF 969–2013) and Human Frontier Science Program Postdoctoral Fellowship (HFSP, LT000627/2014 L) to DEC. A portion of the work was also carried out in the laboratory of DEC. under USDA-NIFA-PBI grant 2018-67013-28492 and the National Science Foundation award no. 1936800. Work in the laboratories of MFS and BPHJT. is supported by the Research Council Earth and Life Sciences (ALW) of the Netherlands Organization of Scientific Research (NWO). BPHJT acknowledges support from the Deutsche Forschungsgemeinschaft (DFG, German Research Foundation) under Germany´s Excellence Strategy – EXC 2048/1 – Project ID: 390686111. The funders had no role in study design, data collection and interpretation, or the decision to submit the work for publication.

## Additional information

### Funding

| Funder | Grant reference number | Author |
| --- | --- | --- |
| Nederlandse Organisatie voor Wetenschappelijk Onderzoek | | Michael F Seidl Bart PHJ Thomma |
| European Molecular Biology Organization | Postdoctoral fellowship EMBO ALTF 969-2013 | David E Cook |
| Human Frontier Science Program | Postdoctoral Fellowship HFSP LT000627/2014-L | David E Cook |
| Deutsche Forschungsgemeinschaft | EXC 2048/1 – Project ID: 390686111 | Bart PHJ Thomma |
| Consejo Nacional de Ciencia y Tecnología | | David E Torres |
| United States Department of Agriculture | 2018-67013-28492 | David E Cook |
| National Science Foundation | 1936800 | David E Cook |

The funders had no role in study design, data collection and interpretation, or the decision to submit the work for publication.

### Author contributions

David E Cook, Conceptualization, Data curation, Formal analysis, Funding acquisition, Validation, Investigation, Visualization, Methodology, Writing - original draft, Project administration; H Martin Kramer, David E Torres, Formal analysis, Investigation, Writing - review and editing; Michael F Seidl, Data curation, Formal analysis, Validation, Investigation, Writing - review and editing; Bart P H J Thomma, Conceptualization, Supervision, Funding acquisition, Writing - original draft, Project administration

### Author ORCIDs

David E Cook (ID) https://orcid.org/0000-0002-2719-4701
Bart P H J Thomma (ID) https://orcid.org/0000-0003-4125-4181

### Decision letter and Author response

Decision letter https://doi.org/10.7554/eLife.62208.sa1
Author response https://doi.org/10.7554/eLife.62208.sa2

# Additional files

## Supplementary files

• Supplementary file 1. Supplementary tables. Table 1. Summary of Transposable elements by Family in core and LS regions. Table 2. Dunns test of pairwise differences for TE Families following Kruskal-Wallis test. Table 3. Summary count of TEs by sub-class. Table 4. Kruskal-wallis test statistic for differences between TE sub-classes for genomic variables. Table 5. P-value results from Conover test and BH multiple testing correction for genomic variables summarized over TE sub-classes. Table 6. Contribution of variables to the first 5 dimensions of PCA. Table 7. Confusion Matrix results for Stochastic Gradient Boosting machine learning of 15 independent training-test predictions of LS and core regions. Table 8. Confusion Matrix results for Random Forest machine learning of 15 independent training-test predictions of LS and core regions. Table 9. GMB LS prediction results for each of the 15 rounds of training and testing. Table 10. RF LS prediction results for each of the 15 rounds of training and testing. Table 11. Contingency tables for observed and expected LS versus core designation for in planta induction. Table 12. Contingency tables for observed and expected LS versus core designation for predicted effectors. Table 13.Contingency tables for observed and expected LS versus core designation for proteins with secretion signal. Table 14. Contingency tables for observed and expected TE elements classified as LS and core in the 3 UMAP Groups. Table 15. Kruskal-wallis test statistic for differences between TE UMAP groups across genomic variables. Table 16. Contigency tables for observed and expected genes classified as LS and core in the 3 UMAP Groups3. Table 17. Kruskal-wallis test statistic for differences between Gene UMAP groups across genomic variables. Table 18. *Verticillium dahliae* isolates used for the presence/absence variation.

• Transparent reporting form

## Data availability

The sequencing data for this project are accessible from the National Center for Biotechnology Information (NCBI) Sequence Read Archive (SRA) under BioProject PRJNA592220.

The following dataset was generated:

| Author(s) | Year | Dataset title | Dataset URL | Database and Identifier |
|---|---|---|---|---|
| Cook DE, Kramer HM, Torres DE, Seidl MF, Thomma BPHJ | 2020 | A unique chromatin profile defines adaptive genomic regions in a fungal plant pathogen | https://www.ncbi.nlm.nih.gov/sra/?term=PRJNA592220 | NCBI Sequence Read Archive, PRJNA592220 |

The following previously published dataset was used:

| Author(s) | Year | Dataset title | Dataset URL | Database and Identifier |
|---|---|---|---|---|
| Faino L, Seidl MF, Datema E, van den Berg GCM, Janssen A, Wittenberg AHJ, Thomma BPHJ | 2015 | ingle-Molecule Real-Time Sequencing Combined with Optical Mapping Yields Completely Finished Fungal Genome | https://www.ncbi.nlm.nih.gov/assembly/GCA_000400815.2 | NCBI Assembly, GCA_000400815.2 |

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
