## [Decision Letter]

**Acceptance summary:**

The genome of the fungal pathogen *Verticillium dahliae* can be partitioned into evolutionary stable core regions and evolutionarily dynamic regions that are repeat-rich, gene-poor, devoid of house-keeping genes, and instead encode genes important for host-pathogen interactions; identifying such "adaptive", lineage-specific portions of pathogen genomes is both interesting in its own right and has implications for agricultural practices. The current study reports a large amount of genetic and epigenetic data that are used to train a model to distinguish core and adaptive regions of the genome, which in turn allows a larger fraction of the genome to be identified as potentially adaptive. The model is validated with a diversity dataset, which confirms that the regions classified as adaptive are more likely to contain lineage-specific sequences. Future work will reveal whether different epigenetic marks contribute to different modes of evolution, or whether they are merely a consequence of different selective forces acting on the different regions.

**Decision letter after peer review:**

Thank you for submitting your article "A unique chromatin profile defines adaptive genomic regions in a fungal plant pathogen" for consideration by *eLife*. Your article has been overseen by Detlef Weigel as the Reviewing and Senior Editor, and three peer reviewers. The following individual involved in review of your submission has agreed to reveal their identity: Brett M Tyler (Reviewer #1).

The reviewers have discussed the reviews with one another and have drafted this decision to help you prepare a revised submission.

As we have judged that your manuscript is of interest, but as described below that additional experiments are required before it is published, we would like to draw your attention to changes in our revision policy that we have made in response to COVID-19 (https://elifesciences.org/articles/57162). First, because many researchers have temporarily lost access to the labs, we will give authors as much time as they need to submit revised manuscripts. We are also offering, if you choose, to post the manuscript to bioRxiv (if it is not already there) along with this decision letter and a formal designation that the manuscript is "in revision at *eLife*". Please let us know if you would like to pursue this option. (If your work is more suitable for medRxiv, you will need to post the preprint yourself, as the mechanisms for us to do so are still in development.)

Summary:

The fungus *Verticillium dahliae* is a broad host-range plant pathogen. As in other filamentous plant pathogens, its genome can be partitioned into evolutionary stable core regions and evolutionarily dynamic regions that are repeat-rich, gene-poor, devoid of house-keeping genes, and instead encode genes important for host-pathogen interactions. Identifying such "adaptive", lineage-specific (LS) portions of pathogen genomes is both interesting in its own right and has implications for agricultural practices.

It has been shown before that these regions (or even entire chromosomes) can exhibit particular epigenetic marks and sequence composition (for example in Fusarium, Zymoseptorira, Leptospheria, Botrytis and others). The current study builds on these observations. It reports an impressive amount of data including whole-genome bisulfite, histone modification, DNA accessibility and RNA-seq data for *V. dahliae*, and these data are used to train a model to distinguish core and adaptive regions of the genome, which in turn allows a larger fraction of the genome to be identified as potentially adaptive. The model is validated with a diversity dataset, which confirms that the regions classified as adaptive are more likely to contain lineage-specific sequences.

What remains to be demonstrated is how the expanded set of "adaptive" regions reported here can be used to advance our understanding of the processes that underlie genome compartmentalization in filamentous plant pathogens, and how different epigenetic marks contribute to different modes of evolution, or whether they are merely a consequence of different selective forces acting on the different regions.

Essential revisions:

1) Too little attention is paid to chromosome organization such as centromeres and pericentromeres, which are linked to differences in epigenetic states.

2) A mutant strain impaired in DNA methylation (Δhp1) is used to show a role of HP1 in DNA methylation in *V. dahliae*. This functional aspect could greatly enhance the study, but is currently poorly integrated in the remainder of the work. It is not clear if the Δhp1 mutant has an overt phenotype or is altered in its gene expression pattern. The Δhp1 mutant must be more fully characterized as to changes in gene expression between core and LS regions.

3) Throughout the manuscript there is reference to repeat induced point mutations (RIP) without comments on the paradox that *V. dahliae* is asexual and that the RIP mechanism is associated with meiosis. Could the signatures of RIP reflect mutations that accumulated in an ancestral sexual population of *V. dahliae*? The absence of RIP mutations from LS regions would seem to be in agreement with these regions having evolved only after the transition to an asexual lifestyle. Also, in agreement, it has previously been shown that TEs associated with LS regions tend to be younger. We ask you to follow up on these previous conclusions.

4) You divide all TEs into four classes (subsection “Transposable element classes have distinct genomic and epigenomic profiles”). Could some patterns be obscured with this broad classification? It appears that the bimodal distributions in Figure 2B reflect different dynamics of different TE (sub)families. Please provide a more fine-scale analysis of properties associated with specific TE (sub)families. Group3 elements are associated with genes that show stronger in planta induction. You should elaborate more on this observation, for example, by describing where in the genome these Group3 elements are located.

5) The machine learning section suffers from a confusion of purpose. You begin by training four algorithms to predict core and lineage-specific (LS) regions from the chromatin data, attempting to minimize false positive and false negatives. Then you pivot, and hypothesize that the remaining false positives may in fact be true positives (previously unclassified LS regions). Validation with new Presence/Absence Variation (PAV) and RNA-seq data supports this proposal. While this approach works, it is not ideal. If you had intended from the beginning to identify new LS regions, you should have created a training data set of high quality core regions (e.g. BUSCO genes and their surrounding regions) as well as high quality LS regions (e.g. genes that encode secreted proteins and that show either PAV or are induced in planta). Furthermore, this curated training data set should contain equal numbers of positive (LS) and negative (core) examples.

6) Following on from the above, in none of the predictions (e.g. in Table 2, Table 3, Figure 6, subsection “Machine learning predicts more lineage-specific genomic regions than previously considered”) do you seem to document which parameter settings were used for each algorithm. As demonstrated in Figure 5, the actual precision and recall vary according to the threshold used to make the binary core/LS call. These threshold parameters should be shown in each case, for example in a diagram similar to Figure 5B or else by showing the optimization curves in the supplement.

Similarly, in the Materials and methods, you state that the optimal parameters for prediction were chosen by maximizing "accuracy". You fail to define "accuracy" but let us assume that you mean [(true positives + true negatives)/all predictions]. This means that, given the heavily skewed distribution of positives and negatives, the optimization will be heavily biased towards minimizing false negatives. In fact, you state "The Matthews correlation coefficient (MCC) [is] an analogous measure to accuracy but more appropriate for unbalanced binary classification". You do not inform us why you used accuracy for the optimizations rather than the MCC, or whether by "accuracy" you intended to convey that you did actually use the MCC.

Here again is confusion of purpose. If you indeed used [(true positives + true negatives)/all predictions], and therefore biased the optimization towards true negatives (core genes) you would inadvertently have liberalized the prediction of false positives. In the absence of a high quality training set, liberalizing the prediction of false positives is actually a good way to improve the search for undetected true positives (LS regions). If this were your intent (which is not discussed), then there would be no need to introduce the MCC except perhaps to explain why it was not used.

7) With this paper, you have substantially expanded the way that LS regions are identified, introducing a much wider range of metrics. Thus, the term "lineage-specific" no longer accurately defines such regions, because it refers only to the use of PAV to define the regions. As you state, you are actually interested in regions responsible for host colonization and adaptation. Continuing to use the term "LS regions" will cause confusion as to whether you refer to regions that exhibit PAV, or regions identified with the new algorithms. This confusion is exemplified by you reference to "old and new LS regions" (subsection “Machine learning predicts more lineage-specific genomic regions than previously considered**”**). We assume that going forward you do not intend to continue to use the terms "old LS" and "new LS". We strongly recommend you choose a new name for the new, expanded "LS" regions.

---

## [Author Response]

Essential revisions:1) Too little attention is paid to chromosome organization such as centromeres and pericentromeres, which are linked to differences in epigenetic states.

We agree that we did not pay much attention to chromosome organization and centromeric phenomena in the original submission. However, just recently we have published a manuscript that characterizes *Verticillium* centromeres in detail (Seidl et al., 2020). Thus, we have now added more information on this topic to the Introduction and Discussion, which also includes integrating the major conclusions from our recent centromere study.

2) A mutant strain impaired in DNA methylation (Δhp1) is used to show a role of HP1 in DNA methylation in V. dahliae. This functional aspect could greatly enhance the study, but is currently poorly integrated in the remainder of the work. It is not clear if the Δhp1 mutant has an overt phenotype or is altered in its gene expression pattern. The Δhp1 mutant must be more fully characterized as to changes in gene expression between core and LS regions.

To provide a more comprehensive analysis of the HP1 mutation, we added information to the Results section as well as Figure 2—figure supplement 1 to address transcriptional changes caused by Δhp1 and specifically in LS versus Core genes. This protein has documented pleiotropic effects, and the exact mechanism leading to differential gene expression in the HP1 mutant is difficult to determine. We have also incorporated findings that are described in another manuscript under review, which describes a more detailed characterization of HP1 (Kramer et al., 2020) in the Results section of our revision. From this work, it is clear that DNA methylation does not control gene expression in *V. dahliae*.

3) Throughout the manuscript there is reference to repeat induced point mutations (RIP) without comments on the paradox that V. dahliae is asexual and that the RIP mechanism is associated with meiosis. Could the signatures of RIP reflect mutations that accumulated in an ancestral sexual population of V. dahliae? The absence of RIP mutations from LS regions would seem to be in agreement with these regions having evolved only after the transition to an asexual lifestyle. Also, in agreement, it has previously been shown that TEs associated with LS regions tend to be younger. We ask you to follow up on these previous conclusions.

Indeed, we did not address the paradox on the observations concerning RIP and the presumed asexual nature of *Verticillium* in the original submission. We agree that the signatures of RIP could reflect mutation that accumulated in an ancestral sexual population of *V. dahliae*, from which asexual lineages sprouted. However, despite the skewed mating type distribution and the abundant occurrence of large-scale genomic rearrangements, we also cannot exclude that local, presently active, sexual populations underly the largely asexual populations. Arguably, in absence of unambiguous proof for the occurrence of sexual activity, this matter cannot be resolved. We have now added a section to the Discussion where we discuss this paradox and its implications.

4) You divide all TEs into four classes (subsection “Transposable element classes have distinct genomic and epigenomic profiles”). Could some patterns be obscured with this broad classification? It appears that the bimodal distributions in Figure 2B reflect different dynamics of different TE (sub)families. Please provide a more fine-scale analysis of properties associated with specific TE (sub)families. Group3 elements are associated with genes that show stronger in planta induction. You should elaborate more on this observation, for example, by describing where in the genome these Group3 elements are located.

We have added further characterization of TE sub-classes for the DNA, LINE and LTR elements to the Results section including Figure 3—figure supplement 1 and Figure 4—figure supplement 1 and Supplementary file 3, Supplementary file 4, and Supplementary file 5. There does not appear to be a clear overall profile on the basis of TE subfamilies that can account for the bimodal distribution.

We believe the question regarding group3 elements and in planta induction is related to the UMAP analysis and Figure 7F. It is actually group1 elements that are the most highly induced in planta, we have updated the y-axis label to more clearly identify the contrast was made as axenic culture over in planta, and so in plant induction is shown as a negative value. As shown in Figure 7D and in the results these genes occur in LS regions far greater than expected by chance.

5) The machine learning section suffers from a confusion of purpose. You begin by training four algorithms to predict core and lineage-specific (LS) regions from the chromatin data, attempting to minimize false positive and false negatives. Then you pivot, and hypothesize that the remaining false positives may in fact be true positives (previously unclassified LS regions). Validation with new Presence/Absence Variation (PAV) and RNA-seq data supports this proposal. While this approach works, it is not ideal. If you had intended from the beginning to identify new LS regions, you should have created a training data set of high quality core regions (e.g. BUSCO genes and their surrounding regions) as well as high quality LS regions (e.g. genes that encode secreted proteins and that show either PAV or are induced in planta). Furthermore, this curated training data set should contain equal numbers of positive (LS) and negative (core) examples.

The reviewers are correct that our original purpose was not to identify new LS regions, but rather to characterize previously identified LS regions in a more comprehensive manner. However, while progressing, we realized that we were able to identify novel LS regions that had previously remained unnoticed. We have added a statement in the Results section to acknowledge this, and highlight that we did indeed pivot the analysis to determining if the “false positives” were indeed newly identified LS regions. As the reviewers note, our results were further validated by independent analysis. As such, despite the approach not being the preferred approach if the initial aim was to identify novel LS regions, we interpret the results as being both novel and providing new biological insight. The suggestions on developing a higher quality training set are valid, and make perfect sense in hind-sight. However, considering our findings and the outcome of the functional validation, it is unlikely that the outcome would be substantially different.

6) Following on from the above, in none of the predictions (e.g. in Table 2, Table 3, Figure 6, subsection “Machine learning predicts more lineage-specific genomic regions than previously considered”) do you seem to document which parameter settings were used for each algorithm. As demonstrated in Figure 5, the actual precision and recall vary according to the threshold used to make the binary core/LS call. These threshold parameters should be shown in each case, for example in a diagram similar to Figure 5B or else by showing the optimization curves in the supplement.

We have now added additional information in the Materials and methods section regarding cross-validation for parameter tuning, assessment, final model parameters and probability thresholds used for binary classification. Additional results from the ML analysis are shown in Figure 6—figure supplement 3 and Figure 6—figure supplement 5.

Similarly, in the Materials and methods, you state that the optimal parameters for prediction were chosen by maximizing "accuracy". You fail to define "accuracy" but let us assume that you mean [(true positives + true negatives)/all predictions]. This means that, given the heavily skewed distribution of positives and negatives, the optimization will be heavily biased towards minimizing false negatives. In fact, you state "The Matthews correlation coefficient (MCC) [is] an analogous measure to accuracy but more appropriate for unbalanced binary classification". You do not inform us why you used accuracy for the optimizations rather than the MCC, or whether by "accuracy" you intended to convey that you did actually use the MCC.

For accuracy, used the standard definition as denoted in the reviewer’s question. We introduced MCC after the ML predictions to help assess model performance. This was because the standard accuracy metric indicated very good model performance (Figure 5A), but we remained skeptical and postulated this was because of the skewed data. The model could have very high accuracy by skewing results to calling most regions negative (i.e. core) generating very high True Negatives and very few False positives. We used accuracy as a metric for model training out of convention, and then later applied the MCC to address the biased results of accuracy to get a better idea of actual model performance. These points are more clearly presented in the Results section of the revision.

Here again is confusion of purpose. If you indeed used [(true positives + true negatives)/all predictions], and therefore biased the optimization towards true negatives (core genes) you would inadvertently have liberalized the prediction of false positives. In the absence of a high quality training set, liberalizing the prediction of false positives is actually a good way to improve the search for undetected true positives (LS regions). If this were your intent (which is not discussed), then there would be no need to introduce the MCC except perhaps to explain why it was not used.

The original intent of the ML analysis was not to identify new LS regions, as we responded above, and have stated in the revision. It is true that our approach may have relaxed predictions in a way that allowed identification of genuine LS regions as evidenced by our analysis. In regards to MCC, it was introduced after running ML to assess model performance. This was because the commonly used metrics, accuracy and auROC, were not informative because of the skewed dataset. MCC provided a more determinative assessment. We have added information to more clearly describe these points in the Results and Discussion of our revision.

7) With this paper, you have substantially expanded the way that LS regions are identified, introducing a much wider range of metrics. Thus, the term "lineage-specific" no longer accurately defines such regions, because it refers only to the use of PAV to define the regions. As you state, you are actually interested in regions responsible for host colonization and adaptation. Continuing to use the term "LS regions" will cause confusion as to whether you refer to regions that exhibit PAV, or regions identified with the new algorithms. This confusion is exemplified by you reference to "old and new LS regions" (subsection “Machine learning predicts more lineage-specific genomic regions than previously considered**”**). We assume that going forward you do not intend to continue to use the terms "old LS" and "new LS". We strongly recommend you choose a new name for the new, expanded "LS" regions.

We have added clearer language in the manuscript where we discuss the old and new LS regions to specify this language is for comparison. We agree with the reviewer that the LS designation may indeed no longer be adequate to describe the regions of interest (i.e. host colonization and adaptation to environment). We have added text in the Discussion to describe the identified DNA as “adaptive genomic regions”.